# High-performance near-infrared OLEDs maximized at 925 nm and 1022 nm through interfacial energy transfer

Chieh-Ming Hung[1,6], Sheng-Fu Wang[1,6], Wei-Chih Chao[1,6], Jian-Liang Li[1], Bo-Han Chen[2], Chih-Hsuan Lu[2], Kai-Yen Tu[1], Shang-Da Yang[2], Wen-Yi Hung[3], Yun Chi[4] ✉ & Pi-Tai Chou[1,5] ✉

Using a transfer printing technique, we imprint a layer of a designated near-infrared fluorescent dye BTP-eC9 onto a thin layer of Pt(II) complex, both of which are capable of self-assembly. Before integration, the Pt(II) complex layer gives intense deep-red phosphorescence maximized at ~740 nm, while the BTP-eC9 layer shows fluorescence at > 900 nm. Organic light emitting diodes fabricated under the imprinted bilayer architecture harvest most of Pt(II) complex phosphorescence, which undergoes triplet-to-singlet energy transfer to the BTP-eC9 dye, resulting in high-intensity hyperfluorescence at > 900 nm. As a result, devices achieve 925 nm emission with external quantum efficiencies of 2.24% (1.94 ± 0.18%) and maximum radiance of 39.97 W sr⁻¹ m⁻². Comprehensive morphology, spectroscopy and device analyses support the mechanism of interfacial energy transfer, which also is proved successful for BTPV-eC9 dye (1022 nm), making bright and far-reaching the prospective of hyperfluorescent OLEDs in the near-infrared region.

The rapid progress of non-fullerene acceptor (NFA) organic photovoltaic (OPV) and deep-tissue bioimaging has led to the vigorous development of organic near-infrared fluorescent dyes. However, for most NIR dyes, the emission efficiency is still quite low[1–7]. This can be explained by the quenching process governed by the emission energy gap law (EEGL, see Eq. (1))[8–11]. For polyatomic molecules, as the emission energy gap decreases, the exciton-vibration coupling between ground and emission states becomes significant, enhancing the radiationless deactivation. One of the key factors to overcome EEGL is to reduce the internal reorganization energy $\lambda$ of the molecule (see Eq. (1))

$$\lambda_{\text{eff}} = \frac{\lambda_{\text{M}}}{N} \tag{1}$$

where $\lambda_{\text{eff}}$ is the effective reorganization energy of the aggregate states, $\lambda_{\text{M}}$ is the reorganization energy of these promoting modes, and $N$ is the exciton delocalization length. In theory, $\lambda_{\text{eff}}$ can be reduced by an orderly molecular assembly that increases exciton delocalization. This can be envisaged with the general knowledge that through adequate molecular packing, single-molecule vibrational displacements can be uniformly distributed through exciton delocalization, effectively reducing the exciton-vibration coupling[12]. On this basis, our team took advantage of heavy metal Pt(II) d⁸ configuration and π-conjugated bi-dentate ligand, forming square planar Pt(II) complexes with non-bonded $d_{z2}$ orbital to facilitate self-assembly. The result not only reduced the energy gap to NIR region by metal–metal-to-ligand charge transfer (MMLCT), but also suppressed the exciton-vibration coupling. This strategy bypasses the shackles of the EEGL and gives

[1]Department of Chemistry, National Taiwan University, Taipei, Taiwan. [2]Institute of Photonics Technologies, National Tsing Hua University, Hsinchu, Taiwan. [3]Institute of Optoelectronic Sciences, National Taiwan Ocean University, Keelung, Taiwan. [4]Department of Materials Sciences and Engineering and Department of Chemistry, City University of Hong Kong, Hong Kong SAR, China. [5]Center for Emerging Materials and Advanced Devices, National Taiwan University, Taipei, Taiwan. [6]These authors contributed equally: Chieh-Ming Hung, Sheng-Fu Wang, Wei-Chih Chao. ✉e-mail: yunchi@cityu.edu.hk; chop@ntu.edu.tw

phosphorescence NIR organic light emitting diode (OLED) with ~1000 nm emission and an external quantum efficiency (EQE) of 4.2%[13].

Considering myriads of organic molecules and their omnidirectional synthetic development, the quest for high-performance fluorescence NIR OLEDs become a goal of our research. In this regard, it is worth noting that a number of NIR emitting dyes have been exploited in bioimaging, which have been reported to enhance fluorescence efficiency at >900 nm upon aggregation in e.g., aqueous media[14–17]. Even so, the photoluminescence quantum yield (PLQY) commonly was reported to be «1%, which is nevertheless sufficient to carry out imaging study because of the low background tissue autofluorescence at >900 nm. However, such low PLQY should not be sufficient in the fabrication of OLEDs. In terms of fluorescence NIR OLEDs, credit should be given to recent work by Xie and co-workers[18], who exploited the V-shaped electron donor–acceptor–donor (A–DA'D–A) dyes that formed a face-on molecular packing. As a result, the fluorescence NIR OLEDs were reported in the region of 900–1000 nm with EQE of 0.15–0.33%. These values, though small, were the world record among fluorescence NIR OLEDs of >900 nm.

The photoluminescent efficiency and charge balance mainly determine the theoretical maximum EQE, which is ~20%[19]. However, the out-coupling efficiency can vary greatly depending on the refractive index of the LED device's multilayers. Thus, it is essential to improve the device structure and materials to increase the out-coupling efficiency. Some recent studies have shown that QD-LEDs and PeLEDs can achieve an EQE above 25% without using any out-coupling enhancement method[20–25].

Herein, taking advantage of the self-assembly and transfer printing, in combination with short-range interfacial energy transfer, we report a leap of fluorescence NIR OLEDs of > 900 nm with a significant increase of both EQE and brightness compared with the current record. The success relies on two self-assembled molecular layers consisting of an energy-donating layer of Pt(fprpz)$_2$ and an energy-accepting layer containing the NIR dyes, such as BTP-eC9 (see Fig. 1). At the beginning, the major obstacle of this approach lies in the impossibility of using vapor-deposited BTP-eC9 in forming the larger framework. Moreover, both mixing Pt(fprpz)$_2$ and BTP-eC9 using co-spin coating and direct spin coating of BTP-eC9 on the top of Pt(fprpz)$_2$ layer failed due to the destruction of the self-assembled Pt(fprpz)$_2$ and BTP-eC9. To overcome this barrier, we found that transfer printing of fluorescent dyes was a suitable solution, which kept the two self-assembled layers intact and allowed the occurrence of interfacial energy transfer efficiently. Although the application of transfer printing in OLED is rare, there are also sporadic reports. However, most reports focus on the transporting layer[26] rather than the emission layer, which is the core of this study. To gain 100% internal conversion

efficiency for the regular NIR fluorescence dye, the exploitation of energy transfer from either triplet or TADF donors is inevitable. As for the emission near NIR(II) region (~1000 nm), up to this stage, only those self-assembled Pt(II) complexes can fulfill the gap for energy transfer. However, the complicated NIR fluorescence dye prohibits vapor deposition. Alternatively, by exploiting the mixing method, the self-assembled structure of the Pt(II) complexes may be destroyed. As a result, the interfacial energy transfer provides a unique solution for the generation of NIR-OLEDs. In this study, we demonstrate two successful arrangements, bi-layered and sandwiched configurations of OLEDs, where Pt(fprpz)$_2$ serves as the energy donor and BTP-eC9 dye acts as the energy acceptor and terminal emitter. For the sandwiched Pt(fprpz)$_2$/BTP-eC9/Pt(fprpz)$_2$ configuration, the device exhibits a peak wavelength at 925 nm with a record high EQE of 2.24% (1.94 ± 0.18%) and maximum radiance of 39.97 W sr$^{-1}$ m$^{-2}$. Importantly, the underlying triplet-to-singlet energy transfer proved to be valid for other NIR dyes, making possible the generation of hyperfluorescent OLEDs in the near-infrared region.

## Results
### Materials selection and their properties
The optimal energy transfer behavior relies upon the donor species exhibiting sufficiently elevated photoluminescence intensity and the acceptor molecule characterized by a high absorption coefficient. It is also imperative that the emission wavelength of the donor overlaps with the absorption spectrum of the acceptor. Additionally, in the energy donor–acceptor (DA) system, the band-gap arrangement should be meticulously designed to conform to the type-I heterojunction-like structure (see Fig. 1c) that effectively mitigates back energy transfer as well as prohibits unwanted electron transfer[27–30]. We opt for Pt(fprpz)$_2$ as the energy donor moiety (Fig. 1a) owing to its capacity to attain a photoluminescence quantum yield (PLQY) surpassing 80% via MMLCT[31]. Pt(fprpz)$_2$ shows remarkable chemical stability and is virtually insoluble in most common organic solvents, enabling surface modifications without compromising intrinsic properties. Moreover, its reduced bandgap of 2.16 eV provides ample options for selecting NIR fluorescence dyes. As the NIR fluorescence dye, we elected to employ BTP-eC9 (Fig. 1b). While another NIR dye Y11, as previously reported by Xie et al.[18], displays a better PLQY, enhanced EQE and stronger emission radiance than that of pure BTP-eC9, in the current study exploiting interfacial energy transfer, Y11 did not have better performance than that of BTP-eC9 in the sandwich OLEDs. This will be discussed later in the section on device performance. Apart from forming a type I arrangement with Pt(fprpz)$_2$ (Fig. 1c), BTP-eC9 belongs to a class of Y-family[32–35] that possess exceptional incident photon-electron conversion efficiency (IPCE).

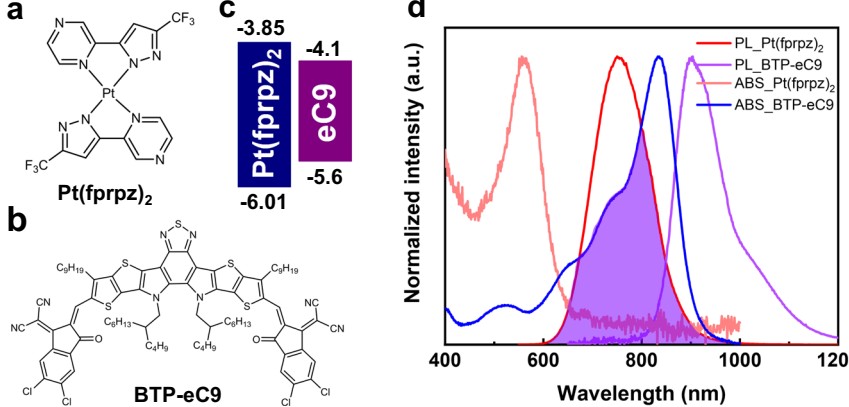

**Fig. 1 | General information on Pt(fprpz)$_2$ and BTP-eC9. a** Chemical structure of Pt(fprpz)$_2$ and **b** BTP-eC9 molecules, along with **c** their energy levels. **d** Absorption and emission spectra of Pt(fprpz)$_2$ and BTP-eC9, with the overlapping region indicating the radiative energy transfer zone.

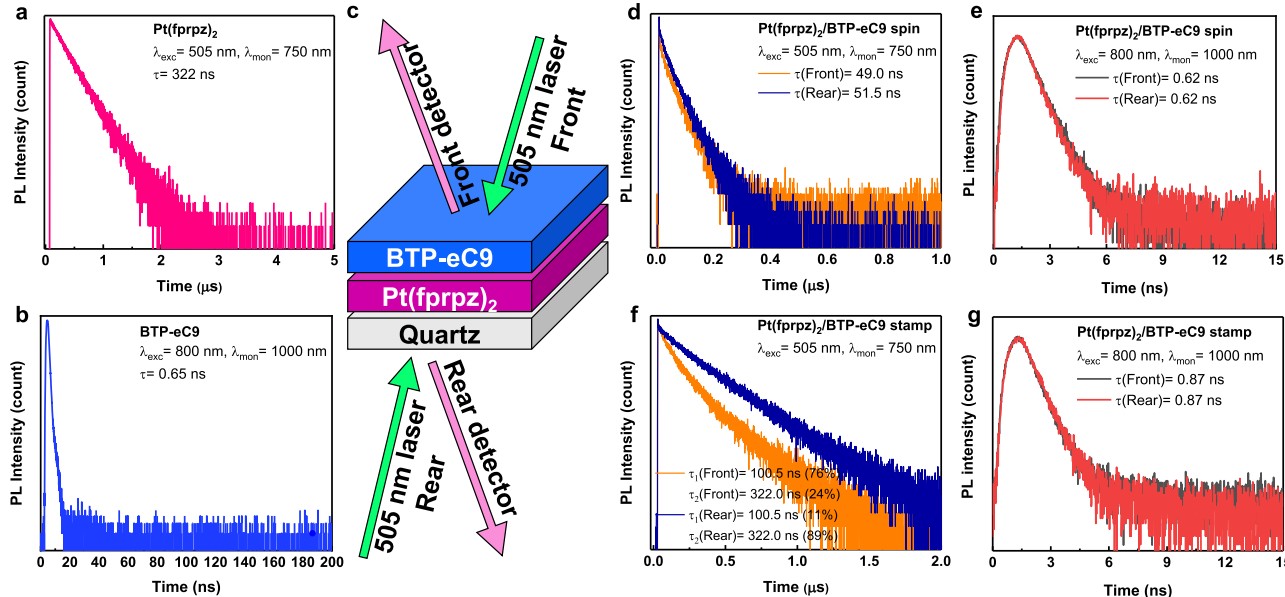

**Fig. 2 | Analysis of time-resolved photoluminescence. a** Pt(fprpz)$_2$ monitored at 750 nm, **b** BTP-eC9 monitored at 1000 nm, **c** We define "front" as the incident light on the surface of BTP-eC9 and "rear" as the incident light on the quartz surface. **d** Pt(fprpz)$_2$/BTP-eC9 spin-coated film monitored at 750 nm, **e** Pt(fprpz)$_2$/BTP-eC9 spin-coated film monitored at 1000 nm **f** Pt(fprpz)$_2$/BTP-eC9 stamped film monitored at 750 nm and **g** Pt(fprpz)$_2$/BTP-eC9 stamped film monitored at 1000 nm. The statistical confidence factor ($\chi^2$) for the lifetime fit is <1.3. The uncertainty of the fitted lifetime value is ±1.2 ns.

The Y-family dyes display a protracted dissociation time at the interface[36]; these properties are unfavorable for organic photovoltaics (OPV) but are highly desirable for OLEDs, rendering an NIR emission at 940 nm in pure solid film. Figure 1d illustrates the absorption spectrum of BTP-eC9, which effectively overlaps with the emission of Pt(fprpz)$_2$, hence facilitating energy transfer. It is worth noting that the radiative decay rate constant for Pt(fprpz)$_2$ phosphorescence is on the order of $10^6 \, s^{-1}$ due to a strong spin–orbit coupling matrix and hence great state mixing between $S_1$ and $T_1$ states. Therefore, the $T_1 \to S_0$ has a large transition dipole, and the Pt(fprpz)$_2$ ($T_1$) → BTP-eC9 ($S_1$) energy transfer can be deemed as a Förster resonance type of energy transfer.

To explore the interlayer properties, we first carried out in-depth analyses using time-resolved photoluminescence (TrPL) with a femtosecond laser excitation at 505 nm where BTP-eC9 absorption is at a minimum. As a result, the single layer of Pt(fprpz)$_2$ exhibited 750 nm emission with a lifetime of 322.0 ± 1.2 ns, and photoluminescence quantum yield (PLQY) was measured to be ~80%, consistent with previously reported findings[31] (see Fig. 2a). In yet another experiment, the single BTP-eC9 layer displayed a significantly shorter lifetime of 0.65 ± 0.04 ns at 1000 nm, coupled with a PLQY of 5.57% (5.42 ± 0.17%) with excitation at 800 nm (refer to Fig. 2b). Prior to delving into the spectral properties of the bilayer emitters, it is essential to establish the configurations of the spectral measurement for the bilayer prepared by BTP-eC9 and Pt(fprpz)$_2$. We specified the configuration as the front when the excitation light first impinged upon the BTP-eC9 layer and rear when the excitation light encountered the quartz substrate that was vapor-deposited by Pt(fprpz)$_2$ (refer to Fig. 2c). In the initial approach, we applied direct spin-coating of BTP-eC9 (in CHCl$_3$) onto the Pt(fprpz)$_2$ layer.

The time-resolved photoluminescence curve of the Pt(fprpz)$_2$/BTP-eC9 spin-coated film monitored at 750 nm is shown in Fig. 2d. Note that four samples were prepared for each experimental condition and measured separately to obtain an average value and error. The decay for front and rear excitation was fitted to be $\tau$(front) = 49.0 ± 1.2 and $\tau$(rear) = 51.5 ± 1.2 ns, respectively, which within the experimental uncertainty, are about the same. However, this value is significantly smaller than the emission lifetime (322.0 ± 1.2 ns) of the pure vapor

deposited Pt(fprpz)$_2$. Furthermore, in the Pt(fprpz)$_2$/BTP-eC9 spin-coated film, upon rear-side excitation where Pt(fprpz)$_2$ was excited directly, the resulting BTP-eC9 920 nm emission was much weaker than that of the front-side excitation where BTP-eC9 was excited (see Supplementary Fig. 1). The results clearly indicate lack of energy transfer from Pt(fprpz)$_2$ to BTP-eC9 in the Pt(fprpz)$_2$/BTP-eC9 spin-coated film. In addition, during the spin coating process of the pre-pared single-layer Pt(fprpz)$_2$ film after being rinsed with chloroform solvent (Supplementary Fig. 2), the emission wavelength of Pt(fprpz)$_2$ blue-shifted from 750 nm to 735 nm, and the PLQY dropped from 78.6% (77.6 ± 0.92%) to 0.92% (0.64 ± 0.24%) (Table S1). The results showed that the assembly change of Pt(fprpz)$_2$ from arrangement to amorphous resulted in a significant decrease in quantum efficiency. We thus conclude that the drastic quenching of Pt(fprpz)$_2$ emission lifetime in the Pt(fprpz)$_2$/BTP-eC9 spin-coated film does not originate from the energy transfer but from the inferior morphology of the Pt(fprpz)$_2$ film that is destroyed by mixing with the BTP-eC9 part upon spin coating. This viewpoint was also confirmed by GIWAXS, AFM, and PL-microscope mapping (vide infra).

The decrease in PLQY of BTP-eC9 after spin-coating onto the Pt(fprpz)$_2$ layer can be attributed to the physical contact between BTP-eC9, solvent, and Pt(fprpz)$_2$ layer at the interface. The different eva-poration rates of the solvent between the two layers of BTP-eC9 and Pt(fprpz)$_2$ will generate thermal stress, leading to the formation of multiple interfaces and seriously damaging the assembly of the molecules. Alternatively, we employ a stamping technique to transfer solid-state BTP-eC9 directly onto the Pt(fprpz)$_2$ layer, thus avoiding unnecessary adverse interfacial interactions (refer to Supplementary Fig. 2 and Supplementary Movie 1; in Supplementary Movie 1, from 0 to 20 s, BTP-eC9 is directly spin-coated onto Pt(fprpz)$_2$, and from 21 to 39 s, BTP-eC9 is stamped onto Pt(fprpz)$_2$ using PDMS). Interestingly, despite its popular exploitation in OPV, the stamping technique was rarely applied in OLEDs[37–41]. Figure 2f shows the bilayer prepared by the stamping method. Due to the efficient energy transfer to BTP-eC9, the emission lifetime of Pt(fprpz)$_2$ is reduced from 322.0 to 100.5 ns, and the energy transfer ratio is calculated to be 68.8%. Therefore, we divide the emission lifetime into two components, where $\tau_1$ (100.5 ± 1.2 ns)

represents energy transfer, and the other is denoted by $\tau_2$ ($322.0 \pm 1.2$ ns) without energy transfer. Their corresponding pre-exponential factor $a_1$ refers to the proportion of energy transfer, and $a_2$ refers to the proportion without energy transfer. By analyzing $a_1$ under different conditions, we can obtain information about where the energy transfer occurs. As a result, $a_1$ of $0.76 \pm 0.04$ deduced from the front direction is significantly higher than $a_1$ of $0.11 \pm 0.02$ detected from the rear side excitation, indicating that most of the energy transfer from Pt(fprpz)$_2$ occurs close to the BTP-eC9 side, i.e. around the interfacial area. Figure 2g shows that the lifetime of the stamped film monitored at 1000 nm is increased to $0.87 \pm 0.02$ ns compared to $0.65 \pm 0.02$ ns of the pure BTP-eC9 film. The PLQY achieved via the stamping method was 8.85% ($8.61 \pm 0.21$%), higher than the PLQY of BTP-eC9. The results clearly indicate that the arrangement of BTP-eC9, after stamped on Pt(fprpz)$_2$, was affected to shorten the d-spacing, thereby improving its face-on orientation[42] and hence PLQY. Relevant steady-state PL, lifetime, and PLQY data based on the stamping method are summarized in Table S1 of supporting information.

## Mechanism of interface energy transfer

The depiction of interfacial energy transfer mechanisms is illustrated in Fig. 3, where the pathway marked in sky blue lines represents the primary process for the interfacial energy transfer. Regarding Pt(fprpz)$_2$, its involvement in metal-to-ligand charge transfer (MLCT) and metal-to-metal-to-ligand charge transfer (MMLCT), facilitated by heavy Pt atom and Pt-Pt interaction, respectively, induces an ultrafast intersystem crossing (ISC) with a rate constant of $k_{isc} > 10^{12}\,s^{-1}$, thereby populating the $T_1$ state with ~100% efficiency. The strong spin-orbit coupling and, hence mixing with the singlet-manifold leads to the $T_1 \rightarrow S_0$ transition virtually allowed, which is evidenced by its sub-microsecond radiative lifetime of phosphorescence[43]. Conversely, the $T_1'$ state of BTP-eC9, being spin-forbidden for the $T_1' \rightarrow S_0'$ transition, necessitates that the energy transfer from the $T_1$ state of Pt(fprpz)$_2$ to the $T_1'$ state of BTP-eC9 employs an electron exchange mechanism, namely the Dexter-type energy transfer. This electron-exchange type energy transfer requires overlap of the electronic wavefunction and hence takes place within a short distance (e.g., <1.5 nm), which should be rather inefficient. Additionally, at the interface, loosely bound singlet (triplet) excitons may form, which subsequently undergo dissociation to form free carriers[44,45]. This process may have a very small branching ratio but cannot be completely neglected, which will

be elucidated in the section on picosecond transient absorption measurement.

## Interface, surface morphology, and carriers dynamics

In a brief summary, the utilization of the stamping method has been found to enhance the photoluminescence quantum yield (PLQY) of BTP-eC9, indicating that both stamping and spin-coating techniques have a significant impact on the morphology and arrangement of Pt(fprpz)$_2$ film. Supplementary Fig. 3 demonstrates that PL-mapping reveals comet-like tails in the spin-coated film, whereas the stamping method maintains a granular morphology. This phenomenon is further confirmed by atomic force microscopy (AFM) (cf. Supplementary Fig. 4). It indicates that BTP-eC9 forms partial erosion during the spin coating process, thereby affecting the integrity of the Pt(fprpz)$_2$ film. It is also reasonable to infer again that the shortened emission lifetime of Pt(fprpz)$_2$ (for Pt(fprpz)$_2$/BTP-eC9 spin-coated film) is mainly due to structural changes rather than quenching caused by energy transfer. Additional evidences were obtained through Kelvin probe force microscopy (KPFM), which measured the surface potential of Pt(fprpz)$_2$ film at 1.03 eV, BTP-eC9 film at 0.77 eV, stamped film at 0.98 eV, and spin-coated film at 1.17 eV. This suggests that spin-coating generates multiple interfaces and defects, leading to charge separation. Grazing-incidence wide-angle X-ray scattering (GIWAXS) analysis, as depicted in Fig. 4a, reveals that Pt(fprpz)$_2$ film tends to exhibit highly ordered edge-on π–π stacking, while BTP-eC9 shows lamellae-type face-on π–π stacking, as shown in Fig. 4b. It is observed that the crystalline signal of Pt(fprpz)$_2$ deteriorates when employing the spin-coating method, indicating physical disruption caused by spin-coating. Therefore, the results presented in Fig. 4c are not surprising. However, Fig. 4d demonstrates that the characteristic peaks of the stamped film remain almost unchanged. Furthermore, integration analysis indicates that the d-spacing of the parallel planes of BTP-eC9 in the stamped film is smaller than that of pure BTP-eC9 (refer to Supplementary Fig. 5), facilitating exciton recombination and hence increases the quantum yield of BTP-eC9 (vide supra). Additionally, angle-dependent near-edge X-ray absorption fine structure (NEXAFS) analysis reveals that the edge-on characteristic peak of the stamped film is blue-shifted by 0.11 eV compared to pure Pt(fprpz)$_2$ film (see Supplementary Fig. 6). The result suggests the formation of T-shaped dimers at the interface between BTP-eC9 and Pt(fprpz)$_2$, enhancing the quadrupole moments[46].

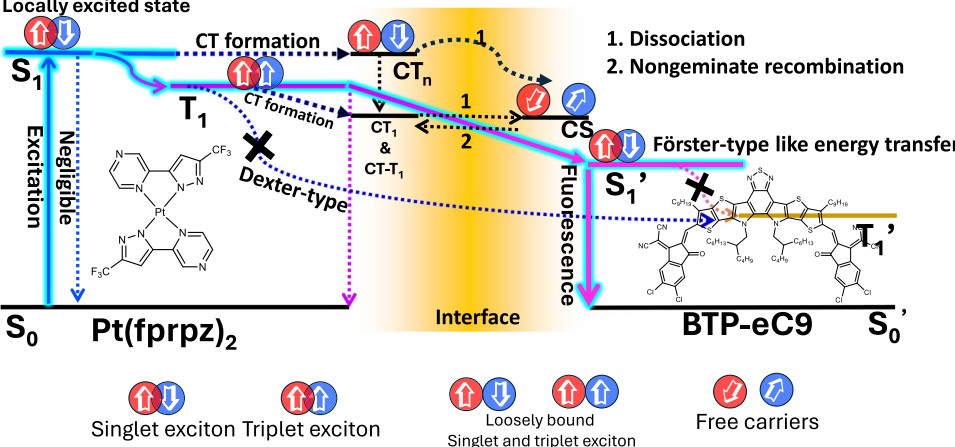

**Fig. 3 | The Jablonski diagram of interfacial energy transfer.** This demonstrates an overview of the role of interfacial energy transfer dynamic processes underlying NIR OLED functionality. The solid sky-blue pathway represents the process of interfacial energy transfer, capable of facilitating FRET. Within this framework, the $S_0$ and $S_1$ states denote the ground and excited states, respectively, in the singlet manifold, while the $T_1$ state represents the triplet state. Note that $T_1 \rightarrow S_1'$ FRET is viable because the $T_1 \rightarrow S_0$ transition is virtually allowed for the Pt (II) complex due to its strong spin-orbit coupling. The diagram also outlines alternative pathways with dashed lines, where charge transfer (CT) and charge transfer-triplet (CT-T) states are included, alongside the charge separation (CS) state, which represents subsidiary processes occurring with a rather small probability.

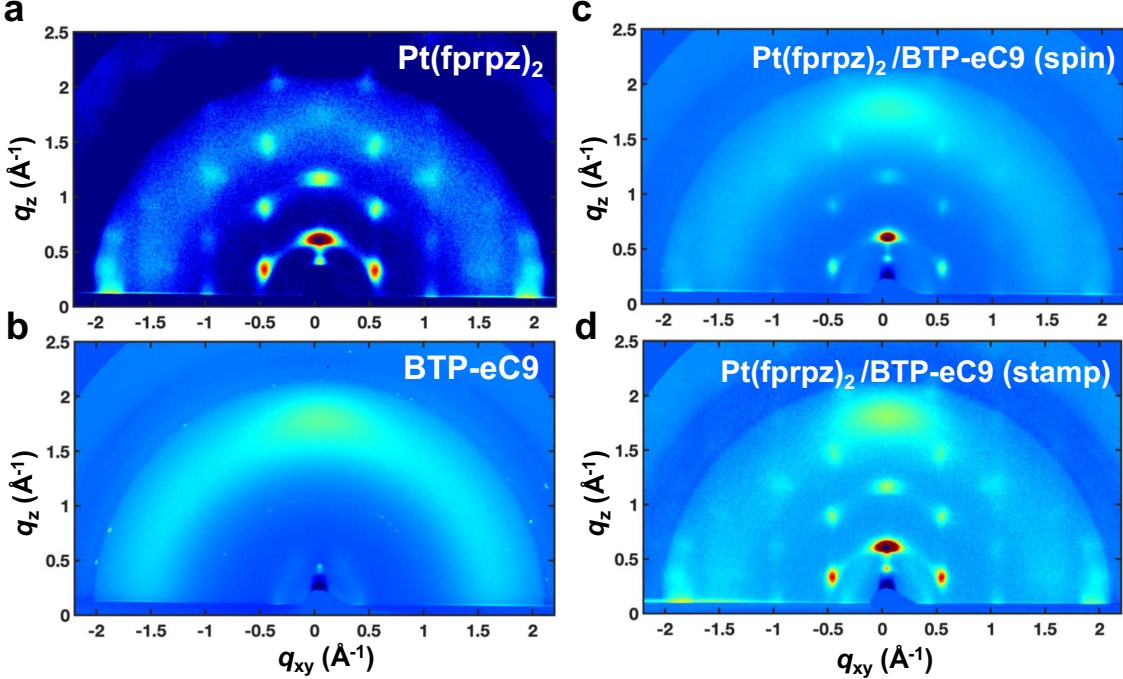

**Fig. 4 | The impact of spin and stamp on molecular packing.** GIWAXS patterns of the studied active layer of **a** Pt(fprpz)$_2$, **b** BTP-eC9, **c** spin-coated Pt(fprpz)$_2$/BTP-eC9, and **d** stamped Pt(fprpz)$_2$/BTP-eC9.

In addition to GIWAXS and AFM measurements, we also explored the carrier behaviors possibly influenced by interfaces via picosecond-transient absorption spectra (ps-TAS). ps-TAS has been widely applied to study the carrier extraction behavior of organic photovoltaics (OPVs), yet research on ps-TAS in organic light-emitting diodes (OLEDs) is scarce. BTP-eC9 belongs to the derivatives of Y-family with the same core chromophore as the emitter, so there is a wealth of literature references to provide. In the ps-TAS study of Y-family derivatives, many reports have identified the 780 nm transient absorbance signal as indicative of the charge separation (CS) signal[45,47,48]. On the other hand, the major transient absorption of the triplet exciton state of BTP-eC9 should be in ~1400 nm region[45]. Therefore, the observed 780 nm transient positive absorbance can be clearly ascribed to the TAS of the CS signal.

We transfer the concept in OPV but opposite consequence to the NIR-bilayer emitters in OLEDs, where the generation of any CS is undesirable, as it would reduce the charge recombination. Upon 650 nm excitation, in pure BTP-eC9, we did not detect any CS signals, indicating the absence of interfaces (see Supplementary Fig. 7). However, when BTP-eC9 was spin-coated onto Pt(fprpz)$_2$, a CS signal emerged at 85 ps (Fig. 5a and c), indicating the creation of certain interfaces due to physical contact. Conversely, when BTP-eC9 was transferred onto Pt(fprpz)$_2$ using a stamping method, the CS signal was delayed to 355 ps (see Fig. 5b and c) with significantly smaller transient absorbance (positive ΔOD, cf. spinning method), suggesting that stamping does not generate a multitude of interfaces. For clarity, Fig. 5d schematically illustrates the exciton dynamics around interfaces prepared by different methods. Here, we must emphasize that the ps-TAS measurement is via optical pumping, which only provides supplementary support for the difference in interface formation between spin coating and imprinting techniques. The real influence of interface structure on the OLED performance should be probed by electric pumping, which is unfortunately not feasible due to the much slower time response[49].

## Electrical measurements and device performance

Prior to the fabrication of NIR-OLEDs, it is essential to gain insight into the charge transport and recombination dynamics. Hence, we analyzed the charge mobilities of the emission layers using the Space charge-limited current (SCLC) technique. For the hole-only device configuration (ITO/MoO$_3$/Emission layers/MoO$_3$/Al), the hole mobility of Pt(fprpz)$_2$ was measured at $3.95 \times 10^{-6}$ cm$^2$ V$^{-1}$ s$^{-1}$, and for BTP-eC9, it was $4.22 \times 10^{-5}$ cm$^2$ V$^{-1}$ s$^{-1}$ (see Supplementary Fig. 8). In the electron-only device configuration (ITO/Al/Emission layers/Al), the electron mobility for Pt(fprpz)$_2$ was found to be $2.67 \times 10^{-5}$ cm$^2$ V$^{-1}$ s$^{-1}$, and for BTP-eC9, it was $1.59 \times 10^{-5}$ cm$^2$ V$^{-1}$ s$^{-1}$ (see Supplementary Fig. 8). Despite the differences in mobility between Pt(fprpz)$_2$ and BTP-eC9, our simulations with Setfos have revealed that the alignment of energy levels between the energy donor and acceptor is more crucial than their mobility. (see Supplementary Figs. S9 and S10).

According to our publication in 2017[31], the Pt(fprpz)$_2$ OLED emitter achieved an EQE of up to 24%. However, the thickness of Pt(fprpz)$_2$ was as high as 20 nm, making such a thick layer unsuitable for interfacial energy transfer. To address this issue, we adjusted the thickness to 10 nm, and the results are shown in Supplementary Fig. 11. The corresponding turn-on voltage was 4.2 V, radiance was 51.95 W sr$^{-1}$ m$^{-2}$, EL was at 720 nm, and EQE was 11.14%. On the other hand, the thickness of BTP-eC9 was ~100 nm, with a turn-on voltage of 1.2 V, radiance of 18.81 W sr$^{-1}$ m$^{-2}$, EL at 952 nm, and EQE of 0.18% (0.14 ± 0.04), as shown in Table 1 and Supplementary Fig. 11.

Next, the Pt(fprpz)$_2$ (10 nm)/BTP-eC9 stamped structure exhibited remarkable improvement, although the turn-on voltage was increased to 6.0 V, it achieved a high radiance of 31.73 W sr$^{-1}$ m$^{-2}$ and an EQE of 2.00% (1.80 ± 0.14). However, we noticed that 11.2% of the emitted light came from Pt(fprpz)$_2$ (Fig. 6c). This may be attributed to its thicker Pt(fprpz)$_2$ layer, which causes some electron holes to recombine within the Pt(fprpz)$_2$ layer. Meanwhile, the redshift of BTP-eC9 emission to 920 nm may be a result of the reduced film thickness. Despite these achievements, we aimed to further suppress Pt(fprpz)$_2$'s emission by optimization of its thickness and used a Setfos software[50,51]

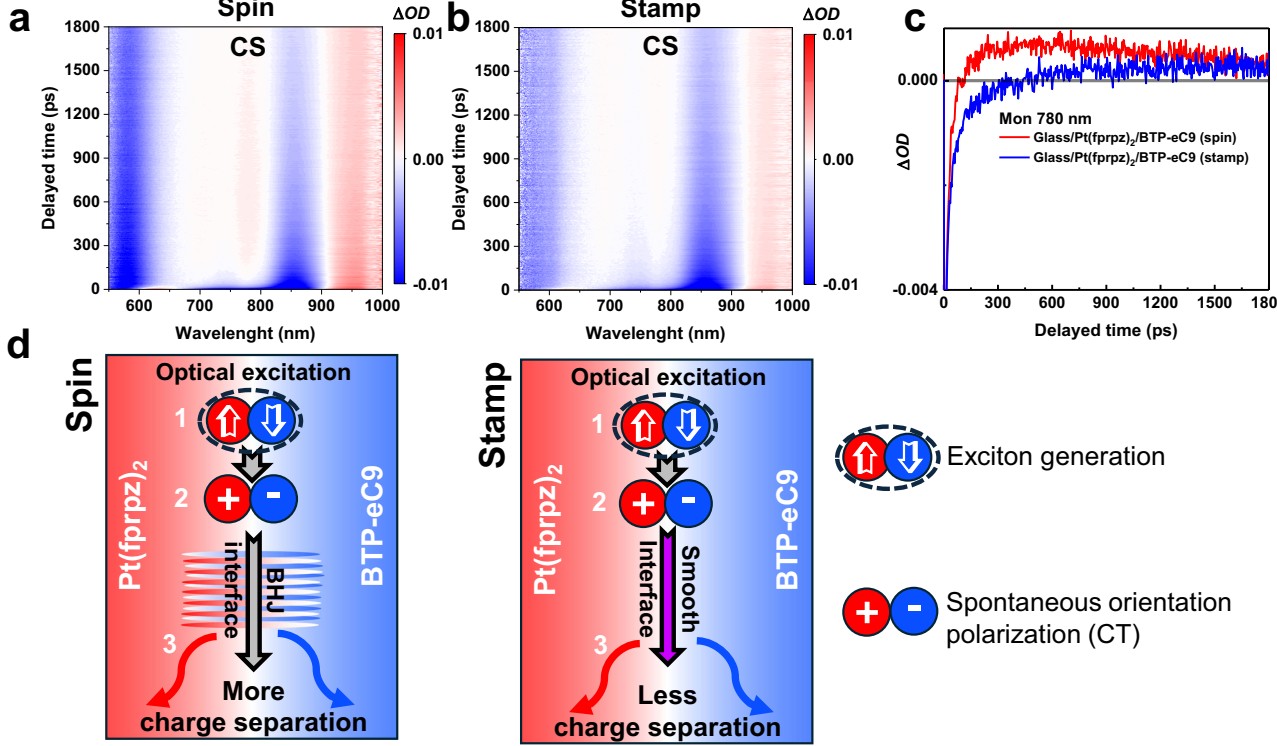

**Fig. 5 | TA results of films treated with spin or stamp methods.** Pseudo-color plots of the picosecond transient absorption (ps-TA) spectra for **a** Pt(fprpz)₂/BTP-eC9 spin-coated film and **b** Pt(fprpz)₂/BTP-eC9 stamped film. **c** The time-dependent transient absorbance (ΔOD) of Pt(fprpz)₂/BTP-eC9 spin-coated and stamped films was monitored at 780 nm. **d** Schematic diagram of exciton dynamics at interfaces prepared by different methods. Process (1): Optically excited exciton generation. Process (2): The spontaneous orientation polarization in the interface facilitates the charge-transfer (CT) formation and (3) subsequent charge separation (CS). Left of **d**: The bulk heterojunction (BHJ) formed due to spin-coating leads to abundant interfaces, providing sufficient driving force for CT formation that undergoes charge separation. Right of **d**: The smooth and plane interface produced by stamping facilitates the production of fewer CT states and, thus, fewer CS separations.

**Table 1 | EL performance of the NIR OLEDs**

| structure | $V_{on}$ (V) | R/J/V (W sr⁻¹ m⁻²/mA cm⁻²/V) | EQE$_{max}$ (%) (average ± error)[a] | $\lambda_{max}$ (nm) |
|---|---|---|---|---|
| Pure BTP-eC9 | 1.3 | 18.81/2686/4.8 | 0.18 (0.14 ± 0.04) | 952 |
| Pt(fprpz)₂(10 nm)/BTP-eC9 | 6.0 | 31.73/312/16.0 | 2.00 (1.80 ± 0.14) | 692 (11.2%), 918 (88.8%) |
| Pt(fprpz)₂(5 nm)/BTP-eC9 | 6.0 | 34.14/306/15.4 | 2.07 (1.83 ± 0.15) | 682 (4.3%), 920 (95.7%) |
| Sandwiched | 6.2 | 39.97/414/17.0 | 2.24 (1.94 ± 0.18) | 682 (3.9%), 925 (96.1%) |

[a] The average EQE is calculated from 16 devices.

to verify this idea. By simulating the recombination area distribution for the EL experiment, EL takes place near the interface between Pt(fprpz)₂ and BTP-eC9 (see Supporting Information). Therefore, the probability of energy transfer is greatly increased because the distance r between Pt(fprpz)₂ and BTP-eC9 is much shorter than the Förster radius $R_0$ calculated to be 7.8 nm (for detail, see Supporting Information). Note that thicker films will cause some electrons and holes to recombine in areas far from the interface. Especially, when $r$ is $>R_0$ the energy transfer probability is greatly reduced, resulting in non-negligible residual Pt(fprpz)₂ emission. The simulation results indicated that as the thickness of Pt(fprpz)₂ decreased, the emission from BTP-eC9 increased (see Supplementary Fig. 12). This postulation was also supported by experiments, as the Pt(fprpz)₂ (5 nm)/BTP-eC9 stamped structure exhibited an increased radiance from 31.73 to 34.14 W sr⁻¹ m⁻² and a slightly improved EQE from 2.00 (1.80 ± 0.14%) to 2.07 (1.83 ± 0.15%), with Pt(fprpz)₂'s emission reduced from 11.2% to 4.3%. We also measured the horizontal dipole ratios, where the Θ value for Pt(fprpz)₂ alone was found to be 78%, indicating the effect of

light out-coupling. In contrast, the Θ value for BTP-eC9 was 67%, and for the Pt(fprpz)₂/BTP-eC9 (stamp) monitored at 950 nm, the Θ value remained at 67%, demonstrating that interface energy transfer does not affect the light out-coupling efficiency (see Supplementary Fig. 14).

Subsequently, through further simulations, we found that using a Pt(fprpz)₂ (5 nm)/BTP-eC9 stamped structure with an additional Pt(fprpz)₂ (5 nm) layer on top (defined as the sandwiched structure) would yield more NIR radiation (see Supplementary Fig. S13). Experimentally, the radiance of the sandwiched structure further increased to 39.97 W sr⁻¹ m⁻², and the EQE reached an unprecedented 2.24% (1.94 ± 0.18%) (Fig. 6). Figure 6a illustrates the luminance and current-voltage characteristics for different structures, Fig. 6b presents the EQE versus current density characteristics, while Fig. 6c and d show the EL spectra and a schematic representation of the optimized device structure. The OLED performance is summarized in Table 1. Then EL spectra at different voltages are shown in Supplementary Fig. 15. A statistical distribution of EQE in 16 studied OLEDs with sandwiched structure was obtained and shown in Fig. 6e. Finally, Fig. 6f illustrates

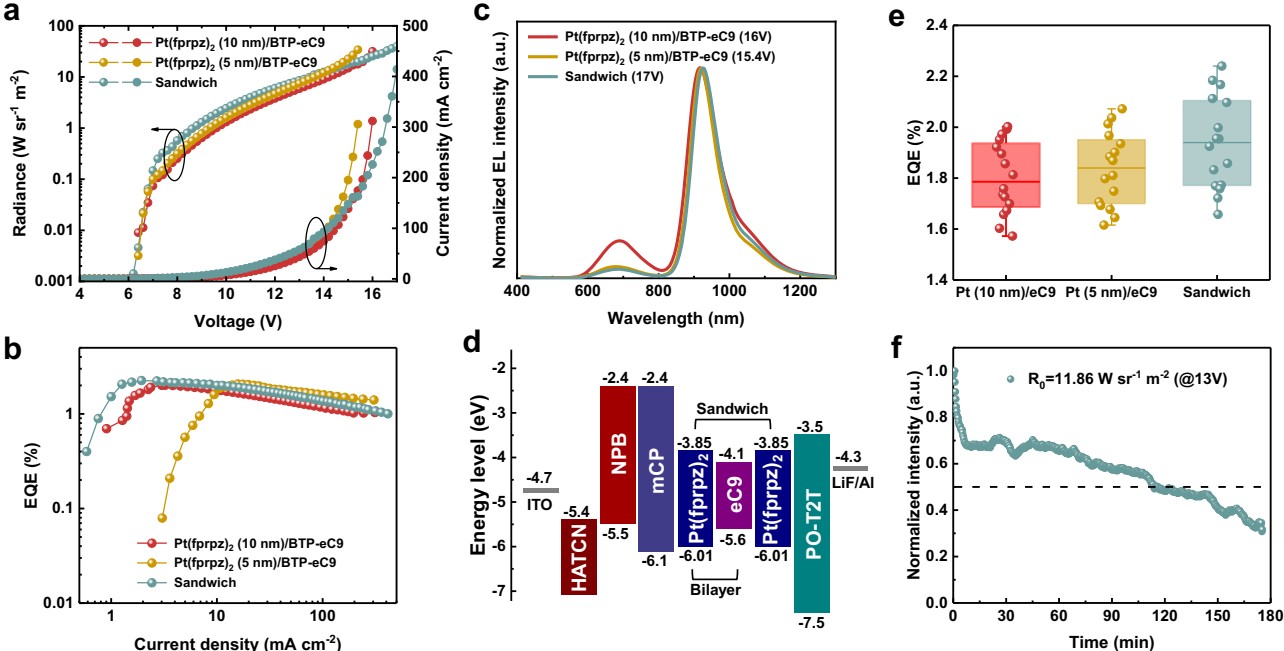

**Fig. 6 | Performance of OLEDs.** The performance of champion device of Pt(fprpz)₂ (10 nm)/BTP-eC9, Pt(fprpz)₂ (5 nm)/BTP-eC9 and sandwich structure. **a** The plot of radiance and current density versus voltages. **b** EQE versus current density curves, **c** EL spectra of Pt(fprpz)₂ (10 nm)/BTP-eC9, Pt(fprpz)₂ (5 nm)/BTP-eC9 in sandwiched structure. **d** Energy level diagram of the studied OLED device. **e** Plot of EQE values based on studied devices. **f** Stability test of the sandwiched devices at 25 °C in ambient air (RH: 50 ± 5 %).

the stability test of the un-encapsulated sandwiched OLED devices recorded at 25 °C in ambient air (with a relative humidity of 50 ± 5%). The initial radiance of 11.86 W sr⁻¹ m⁻² is measured at 13 V, and it takes 103 min to decay to the $T_{50}$ point, indicating the time required for 50% degradation.

The NIR dye Y11 is also studied here, with single-layer dye devices showing enhanced EQE and stronger emission radiance in comparison to that of BTP-eC9. Despite these advantages, however, the LUMO energy level of −3.87 eV for Y11 is nearly in line with that of −3.85 eV for Pt(fprpz)₂, resulting in incomplete charge dispersion at the interface. Supplementary Fig. 16 presents the performance of the Pt(fprpz)₂/Y11/Pt(fprpz)₂ device. Although Pt(fprpz)₂/Y11/Pt(fprpz)₂ exhibits interfacial energy transfer similar to that of Pt(fprpz)₂/BTP-eC9/Pt(fprpz)₂, its device performance is slightly deficient compared to that of Pt(fprpz)₂/BTP-eC9/Pt(fprpz)₂. We have compiled pertinent data in Table S3. We therefore believe that overly similar energy levels may affect the distribution of charges and, thus, the performance of NIR OLEDs.

To substantiate the widespread applicability of interfacial energy transfer in NIR-OLEDs, we further synthesized BTPV-eC9 (with PLQY: 2.53 (2.45 ± 0.08)% measured in neat film), for which the π-conjugation backbone is extended, resulting in an EL spectrum red-shift to the NIR(II) region of 1050 nm. We then selected Pt(II) No. 2[52] (see Scheme of Supplementary Fig. 18) as an energy donor, for which the emission is at 800 nm to maximize the overlap with the absorption of BTPV-eC9 (see Supplementary Fig. 18). As a result, for the OLEDs with only BTPV-eC9 as the emitter, the optimized device exhibits an external quantum efficiency (EQE) of 0.08 (0.06 ± 0.02)% and a maximum radiance of 9.69 W sr⁻¹ cm⁻². Upon using the Pt(II) No. 2/BTPV-eC9 double layers, we again successfully demonstrate interfacial energy transfer with an optimized architecture ITO/HATCN (10 nm)/NPB (50 nm)/mCP (15 nm)/Pt(II) No. 2 (4 nm)/BTPV-eC9/PO-T2T (25 nm)/LiF (1 nm)/Al (120 nm), giving a 1022 nm emission with an EQE of 0.66% (0.55 ± 0.10%) and a maximum radiance of 18.67 W sr⁻¹ cm⁻² (see Supplementary Fig. 19). These results show that the interfacial

energy-transfer method is promising for the application of near-infrared OLEDs.

## Discussion

In summary, using self-assembled Pt(fprpz)₂ and BTP-eC9 as energy donor and acceptor, respectively, we report a leap forward in organic NIR OLEDs through interfacial energy transfer. The implementation of the stamping method is key, which significantly enhances the photoluminescence quantum yield (PLQY) of BTP-eC9 on the Pt(fprpz)₂ film, resulting in an optimized bi-layered structure. Grazing-incidence wide-angle X-ray scattering (GIWAXS) and angle-dependent near-edge X-ray absorption fine structure (NEXAFS) analyses provided valuable insights into the morphological and crystalline characteristics of the stamped film. Furthermore, we verified that stamping has effectively eliminated unnecessary interfaces. We also employed TrPL and TAS to confirm the successful energy transfer from Pt(fprpz)₂ to BTP-eC9 in the donor−acceptor interlayer. Subsequently, through simulation, we further optimized the OLED architecture. Finally, by utilizing the sandwiched structure in the NIR OLED, we achieved a remarkable radiance of 39.97 W sr⁻¹ m⁻² and an exceptional EQE of 2.24% (1.94 ± 0.18%) with an emission peak wavelength of 925 nm. This marks the highest radiance and EQE among all reported fluorescence NIR OLEDs in literature. The significance of this study extends beyond our documented achievements, where the sandwiched structure allows for facile substitution of any other NIR dye without altering the rest of the NIR OLED architecture, greatly reducing the experimental optimization process. Ultimately, we have deduced that the design of an interfacial energy transfer device must satisfy the following criteria: (1) The photoluminescence of the energy donor must overlap with the absorption spectrum of the energy acceptor; the former requires a strong photoluminescence quantum yield (PLQY), and the latter necessitates a high absorption coefficient. (2) For an energy donor and an energy acceptor to function effectively, there must be a sufficient difference between their energy levels. If the LUMO levels overlap, this would lead to the charge being distributed evenly rather than localized

at the interface, resulting in an adverse effect. Similarly, the HOMO levels would exhibit analogous phenomena. (3) In OLEDs, the device must be optimized to bring the electron-hole recombination near the interfacial zone within the effective FRET distance so that energy can be transmitted. We firmly believe that this study represents a crucial breakthrough in the field of fluorescence NIR OLEDs.

## Methods

### Device fabrication of Pt(fprpz)$_2$ OLED

Devices were fabricated on without patterned indium tin oxide (ITO, 150 nm, 10 Ω per square) glass substrates. Before being transferred into a vacuum chamber, ITO substrates were cleaned by ultrasonication in 1% neutral detergent in water, then deionized water, followed by acetone, and finally isopropanol for 20 min each and subsequently dried under a stream of dry nitrogen. The substrates were then undergoing UV−ozone treatment for 20 min. The devices were fabricated in a deposition chamber with a basic pressure of $1 \times 10^{-6}$ torr. Pt(fprpz)$_2$ with an optimized structure of ITO (150 nm)/HATCN (10 nm)/NPB (50 nm)/mCP (15 nm)/ Pt(fprpz)$_2$ (10 nm)/TPBi (55 m)/LiF (1 nm)/Al (120 nm) were fabricated.

### Device Fabrication of BTP-eC9 OLED

The BTP-eC9 OLED was fabricated with the device structure of ITO/ PEDOT:PSS (15 nm)/PVK (30 nm)/BTP-eC9 or Y11 (-100 nm)/C$_{60}$ (5 nm)/BCP (2 nm)/Ag (120 nm). After the UV−ozone treatment for 20 min, PEDOT:PSS spin-coated at 5000 rpm for 30 s on ITO and annealed at 150 °C for 20 min. Then transferred into an N$_2$-filled glove box (<0.1 ppm O$_2$ and H$_2$O). PVK (10 mg mL$^{-1}$ in CB) was deposited onto the PEDOT:PSS films with a spin-coating condition of 2000 rpm for 30 s, then annealed at 150 °C for 10 min. After the film cooled down to room temperature, BTP-eC9 or Y11(16 mg mL$^{-1}$ in CF) was in the spin-coating condition of 4000 rpm with the hot solution. 5 nm C$_{60}$, 2 nm BCP, and 120 nm silver electrode were evaporated under high vacuum (<$1 \times 10^{-6}$ Torr) sequentially.

### PDMS stamps

The PDMS stamps were prepared by casting a mixture of the elastomer and the curing agent (10:1). After mixing, the blend was poured onto a silicon wafer to achieve a smooth stamp surface; vacuuming/de-vacuuming was performed for 20 min to remove any air bubbles. The mixture was placed in an oven and heated at 90 °C for 90 min. The cured PDMS stamp was removed from the mold and cut into shapes of appropriate dimensions. For the solution to form a film on PDMS, the PDMS must be treated with UV−ozone for 30 min.

### Device Fabrication of Pt(fprpz)$_2$/BTP-eC9 stamp OLED

The fabrication process involves the sequential deposition on ITO glass of HATCN (10 nm)/NPB (50 nm)/mCP (15 nm)/Pt(fprpz)$_2$ (5–10 nm). Subsequently, BTP-eC9 or Y11 (-35 nm) is spin-coated onto PDMS, which has undergone UV-ozone treatment. It is crucial to note that the transfer quality of PDMS directly influences the performance of the OLED. Finally, Pt(fprpz)$_2$ (5 nm for sandwich structure)/PO-T2T (30 nm)/LiF (1.5 nm)/Al (150 nm) is deposited through thermal evaporation.

### Characterization

The current density−voltage−radiance and Electroluminescence spectra characteristics were characterized by the LQ-50X system (Enlitech, Taiwan) includes a PTFE integrating sphere, a Multi-Channel Photon Detector (MCD) and two spectrometers to collect emission photons and subsequent spectral analyses. The MCD enhances sensitivity, facilitating effective detection in low-light conditions. The system is capable of measuring a broad wavelength range from 300 to 1700 nm using Si and InGaAs detectors, and it is calibrated against a NIST-traceable standard lamp. equipped with a source meter (Keithley

2400). photoluminescence (PL) mapping images were acquired on an LSM 880 (Zeiss). Grazing incidence wide-angle X-ray scattering (GIWAXS) patterns were conducted on beamline BL13A1, and NEXAFS spectra were conducted on beamline BL20A1 in the National Synchrotron Radiation Research Center (NSRRC), Taiwan. The scattering patterns were collected on a Mar165 CCD with a diameter of 40 mm. The scattering vector, $q = 4\pi/\lambda \sin \theta$, along with the scattering angles $q$ in these patterns, were calibrated using silver behenate. Tapping mode atom force microscopy (TM-AFM) and Kelvin probe force microscope (KPFM) images were acquired on a Dimension Icon AFM (Bruker). The absorption spectra were obtained on a Hitachi UH-5700 spectrophotometer. Steady-state PL spectra and time-resolved studies were performed using a time-correlated single photon counting (TCSPC) system (FLS 980, Edinburgh). The valance band of Pt(II) No.2 was confirmed using ultraviolet photoemission spectroscopy (UPS) measurements (ULVACPHI, Japan). $^1$H and $^{13}$C NMR spectra were recorded on AVIII-500 MHz spectrometers, and chemical shifts were measured in $\delta$ (ppm) with residual solvent peaks as internal standards (CDCl$_3$, $\delta$7.26 ppm in $^1$H NMR, $\delta$ 77.00 ppm in $^{13}$C NMR). High-resolution matrix-assisted laser ionization (HR-MALDI) spectra were obtained on a Bruker, New ultrafleXtreme™, Bremen, D.E.

## Data availability

Complete experimental procedures and compound characterization data are available in the Supplementary Information; any other data are available from the authors on request. Source data are provided with this paper.

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

## Acknowledgements

P.-T.C. acknowledges the financial support through the National Science Council of Taiwan (grant no. NSTC 112-2639-M-002-007-ASP). We are grateful to Shu-Chih Haw of the National Synchrotron Radiation Research Center. for the NEXAFS analysis. We thank Dr. Hsi-Ching Tseng at the NTU Instrumentation Center for the assistance in NMR experiments. Mass spectrometry analysis was performed by the Mass Spectrometry facility of the Institute of Chemistry, Academia Sinica, Taiwan.

## Author contributions

C.-M.H. spearheaded the entire experimental process, including the preparation of OLEDs, and authored the initial draft. S.-F.W., J.-L.L., and Y.C. synthesis of Pt(fprpz)$_2$, Pt(II) No. 2 and BTPV-eC9. W.-C.C., K.-Y.T., W.-Y.H., and P.-T.C. conducted optical measurements and Setfos calculated. B.-H.C., C.-H.L., and S.-D.Y. conducted TA measurements. All authors discussed the results and contributed to the paper.

## Competing interests

The authors declare no competing interests.
