## [Peer Review File · Nature Communications]

High-Performance Near-infrared OLEDs Maximized at 925 nm and 1022 nm through Interfacial Energy TransferREVIEWER COMMENTS

Reviewer #1 (Remarks to the Author):

Hung, Wang, Chao et al. present a big advance in performance for near-IR OLEDs using Pt-based energy donor and organic energy acceptor and emitting component, reaching 2.24% EQE at 924 nm. The organic emitting component is BTP-eC9 – has excellent optical properties for near IR emission – can not be vacuum deposited and therefore presents an engineering challenge for how to use these materials in OLEDs. The energy transfer device employs Pt(fprpz)₂ as the sensitizer. The authors therefore employ solution processing methods of spin coating and stamping for the BTP-eC9 layer, and use transient optical spectroscopy to demonstrate they have set up an energy transfer system. They then employ this approach to show the potential for high performance near-IR OLEDs. In my opinion the breakthrough is quite specific to this class of systems (BTP-eC9) is not strictly the only energy acceptor type that could be used in such energy transfer devices. A stronger emphasis on wider applicability – or understanding why BTP-eC9 type emitters is the only way to go – is necessary to be of general interest for the readership of Nature Communications. I am also not confident in the photophysical interpretations of intermediate states (such as exciton dissociated) that they have presented in the paper. For these reasons I do not recommend publication and highlight the following particular issues:

- Line 186 – attributing the signals to exciton dissociated states seems speculative. Are these charge transfer states at the interface? How have the authors ruled out the triplet exciton state of BTP-eC9? The key proposed mechanism that underpins their paper is not supported by the current experimental data.

- The suggestion of an exciton dissociated state should be probed by temperature dependence to understand the energetics in their proposed Jablonski diagram.

Reviewer #2 (Remarks to the Author):

In the work “A Leap Forward in Organic Fluorescent Near-infrared OLEDs through Interfacial Energy Transfer: Peak Wavelength 925 nm, External Quantum Efficiency 2.24%, Radiant Brightness 39.97 W sr⁻¹ m⁻²”, Chieh-Ming Hung and co-authors report OLEDs based on a NIR fluorescent emitter (BTP-eC9) sandwiched between highly efficient phosphorescent Pt complex (Pt(fprpz)₂) layers. The electroluminescence efficiency is the highest ever reported from OLEDs containing a heavy-metal-free active layer emitting above 900 nm. This result is made possible by the efficient interfacial

energy transfer from the Pt complex to the NIR fluorescent dye, which the authors optimised – by depositing the NIR dye via transfer printing, guided by simulations and structural/morphological characterization – and characterised optically.

Some revisions are needed, as detailed in the comments below:

1. In the abstract, line 18, authors state that “...a thin layer of a Pt(II) complex, PtPt(fprpz)₂, that was vapor-deposited on a ITO substrate”. However, the architecture of the final OLED is more complex and interlayers between PtPt(fprpz)₂ and ITO are present. Therefore, the sentence should be amended accordingly.

2. In the introduction, line 55, authors point out that “Despite the above success, the unique features for the Pt(II) complex may impose the limitation in both breadth and versatility of applications due to the required precious Pt(II) element and specific ligands”. I do agree with this, however, OLEDs reported in the present manuscript still include the above complexes. Authors should therefore expand a bit more on the advantages offered by the use of a fluorescent emitter, while still integrating Pt(II) complexes as triplet harvesting materials.

3. Line 134: “The decline of BTP-eC9 PLQY upon its spin coating onto Pt(fprpz)₂ layer is attributed to the chemical interaction between BTP-eC9 and Pt(fprpz)₂ layers in the interface”. Authors should clarify what they mean by “chemical interaction”.

4. Line 141: “...stamping technique was rarely applied in OLEDs”. Please provide at least one of such references on transfer printed OLEDs.

5. Line 150: “The PLQY achieved via the stamping method was 8.85%, higher than the PLQY of BTP-eC9. This enhancement shows that in addition to the energy transfer from Pt(fprpz)₂ to the BTP-eC9 pathway, there are other channels, such as the reabsorption of Pt(fprpz)₂ emission by BTP-eC9”. However, in the Supplementary info file, Table S1, 800 nm excitation wavelength is reported. With this excitation wavelength, no excitation and consequent reabsorption of the Pt(fprpz)₂ emission should occur. In the same Table S1, emission at 754 nm from the complex is also reported, which should not be observed unless energy upconversion is taking place. Authors should clarify this.

6. Figure 2: There seems to be a typo in the legends. λ_{em} should be λ_{exc} , i.e., excitation wavelength.

Reviewer #3 (Remarks to the Author):

This paper reports an interesting NIR OLED result employing sensitized emission of a fluorescent molecule by a sensitizing Pt phosphor. The best results are obtained by stamping of one layer onto another forming a bilayer.

Overall, the engineering results are of interest: I am not aware of such a high efficiency from an OLED at such a long wavelength, although it is much inferior to performance of inorganic LEDs whose IQE approach 100%.

There are numerous deficiencies in the paper that need to be pointed out. There are errors in language and terminology as well.

1. The title is hype. Please tone it down. There is too much claimsmanship in papers already so we should be more modest and let the readers decide whether this is a 'leap forward' or not.
2. Line 96: this is a type I heterojunction, not a quantum well.
3. There are no PL spectra of the mixed or layered structures. This makes it impossible to evaluate the other spectra within the paper.
4. There are almost no error bars to the data. The only error noted is in Fig. 5e, but ALL errors to all measured values should be quoted in the abstract and throughout the text. Only then can I determine whether 51.5 ns and 49 ns in Line 128 are significantly different.
5. Discussions of preexponential factors a_1 and a_2 (paragraph starting on line 139) are unclear. Error bars needed. They need to be rigorously defined.
6. What is the Forster transfer radius? Is this consistent with the drop in efficiency? Also, the Pt diffusion length is needed for complete analysis of performance. But one thing is apparent: There are significant losses in this device since the Pt efficiency is far greater than the final device efficiency. Where do these losses originate?
7. Figures are disorganized. Putting section (c) after (e) is confusing in Fig. 2.
8. The GIWAXS analysis is not useful without knowing the crystal structures. There is reference to d-spacings and stacking habits but without the crystal structure this is not a verifiable assertion.
9. This is not the first demonstration of stamped combinations of organic layers to form a device (or even an OLED). References to the literature are required.
10. I don't know what a negative TAS is due to. Simply stating it is due to free charge, chemical contact, etc. is insufficient. Scientific rigor is missing throughout the paper.
11. There is considerable residual Pt emission. Why? This goes back to my point #6.

12. No analysis is provided to explain the high voltage operation. This needs to be included.

13. What was the set up to measure OLED EQE? Is it within an integrating sphere, or in the forward direction? What detector was used, how was it calculated?

This is a sampling of many more comments and questions that I have. At a minimum, these need to be answered quantitatively and thoroughly.

Reviewer #4 (Remarks to the Author):

Hung and co-workers reported a strategy for fabricating NIR-OLEDs relying on an interfacial energy transfer process. Specifically, the authors took advantages of the superior NIR emissions of self-assembled Pt(II) complex and a fluorescence dye (BTP-eC9), which can be maintained using the transfer-printing method. It is interesting to find that the emission of BTP-eC9 can be enhanced in stamped films probably due to changes in its packing arrangement. The authors performed time-resolved photoluminescence, transient absorption and other spectroscopic measurements to characterize the structure and energy transfer process. The device efficiencies were improved by ca. 6-folds in comparison with the fluorescent OLEDs. This work demonstrates a way to advance the efficiency of NIR OLEDs using fluorescent dye as the terminal emitter. However, there are some issues that need to be addressed before its consideration for publication on Nature Communications.

Major revisions:

1. The interfacial energy transfer mechanism is reminiscent of phosphorescence sensitized fluorescence OLEDs by co-doping the metal complexes and fluorescent emitter. As stated in the manuscript, the authors tried to co-spin coat the mixing Pt(II) complex and BTP-eC9, which was unsuccessful. Therefore, the self-assembled Pt(II) structure is still one of the keys to the achievement in this work, which can effectively suppress exciton-vibrational coupling. From this point of view, it is hard to accept that interfacial energy transfer is really the most or only important factor. I suggest the authors to reorganize the writing with these two considerations: (1) which factor is the most important for NIR-OLEDs using the interfacial energy transfer method? (2) What is the advantage of this interfacial energy transfer method?

2. Following Comment 1, in Line 56, the authors pointed out the limitation of using precious Pt element for NIR OLEDs based on Pt(II) aggregates with triplet MMLCT excited states. This is true. But, for the present interfacial energy transfer strategy, Pt(II) is still required and key to the device

performance. So, it might be not acceptable to attract readers by using “Fluorescent Near-infrared OLEDs”. I suggest the authors to revise the writing in this aspect.

3. In Lines 72-74, the authors remarked the QD-LEDs and PeLEDs with EQEs >25. There have many reports on OLEDs having high EQEs even close to 40%. Preferred horizontal transition dipole moment has been identified as a key parameter dictating the device EQEs. Because the present work is on NIR OLEDs, references should not be only limited to QD-LEDs and PdLEDs. On the other hand, are the self-assembled layers beneficial for increasing the light out-coupling efficiency?

4. The energy transfer from Pt(fprpz)₂ to BTP-eC9 is described as efficient throughout the text. But there are significant portion of emission from the Pt(II) donor in the photoluminescence test. In host-guest system, the efficient energy transfer is usually characterized by a complete attenuation of the donor emission. Although the lifetime is reduced from 322 ns to 100.5 ns, this may not be used as an indicator for this judgement. Also, the device shows emissions from both the Pt(II) and BTP-eC9, revealing that the tuning of energy transfer is not very successful. Although the authors tried to optimize the device structures according to simulation results, they have not tackled this problem. This is very important for evaluating this interfacial energy transfer concept for the design of NIR OLEDs.

5. The authors should elaborate more on the formation of the exciton dissociated (ED) states at the interfaces. For example, how would the band alignment affect this process?

6. Discussion on the charge transport and recombination in devices is missing. With two or three emission layers, the recombination zones should be crucial to the electroluminescence spectra and efficiencies.

Minor revisions:

1. Transfer printing in OLEDs has not been widely explored. I suggest the authors to add key results/progresses on this topic in the Introduction.

2. Please indicate the methods for determining the MO levels for both Pt(II) and BTP-eC9, including their states in either solution or film.

3. In Line 143: “.....the energy transfer rate is calculated to be 68.8%.” I think this data means energy transfer ratio.

REVIEWER COMMENTS

Reviewer #1 (Remarks to the Author):

Hung, Wang, Chao et al. present a big advance in performance for near-IR OLEDs using Pt-based energy donor and organic energy acceptor and emitting component, reaching 2.24% EQE at 924 nm. The organic emitting component is BTP-eC9 – has excellent optical properties for near IR emission – cannot be vacuum deposited and therefore presents an engineering challenge for how to use these materials in OLEDs. The energy transfer device employs Pt(fprpz)₂ as the sensitizer. The authors therefore employ solution processing methods of spin coating and stamping for the BTP-eC9 layer, and use transient optical spectroscopy to demonstrate they have set up an energy transfer system. They then employ this approach to show the potential for high performance near-IR OLEDs. In my opinion the breakthrough is quite specific to this class of systems (BTP-eC9) is not strictly the only energy acceptor type that could be used in such energy transfer devices. A stronger emphasis on wider applicability – or understanding why BTP-eC9 type emitters is the only way to go – is necessary to be of general interest for the readership of Nature Communications. I am also not confident in the photophysical interpretations of intermediate states (such as exciton dissociated) that they have presented in the paper. For these reasons I do not recommend publication and highlight the following particular issues:

Reply: We are thankful to the reviewer for his/her valuable comments; especially the emphasis in that BTP-eC9 should not be the only one to claim the generalization of the mechanism of interfacial energy-transfer for NIR OLEDs. In this revised version, we further prove the concept by using another NIR fluorescence molecule, namely BTPV-eC9, to extend the electroluminescence far to the NIR(II) region of 1050 nm. Detail is elaborated in the following reply and the revised manuscript. The results should affirm the wider applicability of the interfacial energy transfer especially in the NIR region.

- Line 186 – attributing the signals to exciton dissociated states seems speculative. Are these charge transfer states at the interface? How have the authors ruled out the triplet exciton state of BTP-eC9? The key proposed mechanism that underpins their paper is not supported by the current experimental data.

Reply: We apologize for the inappropriate interpretation of the data of picosecond-transient absorption spectra (ps-TAS) and, as a result, have reinterpreted the ps-TAS

results in the revised manuscript, which is also attached below for the reviewer's convenience.

“In addition to GIWAXS and AFM measurements, we also explored the carrier behaviors possibly influenced by interfaces via picosecond-transient absorption spectra (ps-TAS). ps-TAS has been widely applied to study the carrier extraction behavior of organic photovoltaics (OPVs), yet research on ps-TAS in organic light-emitting diodes (OLEDs) is scarce. BTP-eC9 belongs to the derivatives of Y-family with the same core chromophore as the emitter, so there are a wealth of literature references to provide. In the ps-TAS study of Y-family derivatives, many reports have identified the 780 nm transient absorbance signal as indicative of charge separation (CS) signal ^[43-45]. On the other hand, the major transient absorption of the triplet exciton state of BTP-eC9 should be in ~1400 nm region ^[43]. Therefore, the observed 780 nm transient positive absorbance can be clearly ascribed to the TAS of the CS signal.

We transfer the concept in OPV but opposite consequence to the NIR-bilayer emitters in OLEDs, where the generation of any CS is undesirable, as it would reduce the charge recombination. Upon 650 nm excitation, in pure BTP-eC9, we did not detect any CS signals, indicating the absence of interfaces (see Figure S6). However, when BTP-eC9 was spin-coated onto Pt(fprpz)₂, a CS signal emerged at 85 ps (Figure 4c), indicating the creation of certain interfaces due to physical contact such as van der Waal force, hydrogen bonding, π -stacking, etc. in between. Conversely, when BTP-eC9 was transferred onto Pt(fprpz)₂ using a stamping method, the CS signal was delayed to 355 ps (see Figure 4c) with significantly smaller transient absorbance (positive ΔOD , cf. spinning method), suggesting that stamping does not generate a multitude of interfaces. For clarity, Figure 4d schematically illustrates the exciton dynamics around interfaces prepared by different methods. Here, we must emphasize that the ps-TAS measurement is via the optical pumping, which only provides supplementary support for difference of interface formation between spin coating and imprinting techniques. The real influence of interface structure to the OLEDs performance should be probed by electric-pumping, which is unfortunately not feasible due to the much slower time response ^[46].”

Figure 4. Pseudo-color plots of the picosecond transient absorption (ps-TA) spectra for (a) $\text{Pt}(\text{fprpz})_2/\text{BTP-eC9}$ spin-coated film and (b) $\text{Pt}(\text{fprpz})_2/\text{BTP-eC9}$ stamped film. (c) The time-dependent transient absorbance (ΔOD) of $\text{Pt}(\text{fprpz})_2/\text{BTP-eC9}$ spin-coated and stamped films monitored at 780 nm. (d) Schematic diagram of exciton dynamics at interfaces prepared by different methods. Process (1): Optically excited exciton generation. Process (2): The spontaneous orientation polarization in the interface facilitates the charge-transfer (CT) formation and (3) subsequent charge separation (CS). Left of (d): The bulk heterojunction (BHJ) formed due to spin-coating leads to abundant interfaces, providing sufficient driving force for CT formation that undergoes charge separation. Right of (d): The smooth and plane interface produced by stamping facilitates produce less CT states and thus fewer CS separations.

- The suggestion of an exciton dissociated state should be probed by temperature dependence to understand the energetics in their proposed Jablonski diagram.

Reply: We are thankful to the reviewer for the valuable feedback. In response to the previous query, we have corrected our original assignment from "exciton dissociated states" to the state associated with charge separation (CS). Again, we are sorry for the misinterpretation.

Reviewer #2 (Remarks to the Author):

In the work "A Leap Forward in Organic Fluorescent Near-infrared OLEDs through

Interfacial Energy Transfer: Peak Wavelength 925 nm, External Quantum Efficiency 2.24%, Radiant Brightness 39.97 W sr⁻¹ m⁻²”, Chieh-Ming Hung and co-authors report OLEDs based on a NIR fluorescent emitter (BTP-eC9) sandwiched between highly efficient phosphorescent Pt complex (Pt(fprpz)₂) layers. The electroluminescence efficiency is the highest ever reported from OLEDs containing a heavy-metal-free active layer emitting above 900 nm. This result is made possible by the efficient interfacial energy transfer from the Pt complex to the NIR fluorescent dye, which the authors optimised – by depositing the NIR dye via transfer printing, guided by simulations and structural/morphological characterization – and characterised optically.

Some revisions are needed, as detailed in the comments below:

1. In the abstract, line 18, authors state that “...a thin layer of a Pt(II) complex, Pt(fprpz)₂, that was vapor-deposited on a ITO substrate”. However, the architecture of the final OLED is more complex and interlayers between Pt(fprpz)₂ and ITO are present. Therefore, the sentence should be amended accordingly.

Reply: Thanks to the reviewer for his/her valuable suggestions. As suggested by the reviewer, the sentence has been revised as follows (see also revised text in page 1, line 19)

“Using a transfer printing technique, we successfully imprinted a layer of a designated near-infrared (NIR) fluorescent dye BTP-eC9 onto a thin layer of a Pt(II) complex, Pt(fprpz)₂ that was vapor-deposited on an ITO/HATCN/NPB/mCP substrate.”

2. In the introduction, line 55, authors point out that “Despite the above success, the unique features for the Pt(II) complex may impose the limitation in both breadth and versatility of applications due to the required precious Pt(II) element and specific ligands”. I do agree with this, however, OLEDs reported in the present manuscript still include the above complexes. Authors should therefore expand a bit more on the advantages offered by the use of a fluorescent emitter, while still integrating Pt(II) complexes as triplet harvesting materials.

Reply: We are thankful to the reviewer for his/her valuable comments; especially the emphasis in that BTP-eC9 should not be the only one to claim the generalization of the

mechanism of interfacial energy-transfer for NIR OLEDs. During the revision, we further proved the concept by using another NIR fluorescence molecule, namely BTPV-eC9, to extend the electroluminescence far to the NIR(II) region of 1050 nm. The results should affirm the wider applicability of the interfacial energy transfer especially in the NIR region. We have included these results in the revised manuscript (see also below) with the device performance displayed in Figure S13:

“To substantiate the widespread applicability of interfacial energy transfer in NIR-OLEDs, we further synthesized BTPV-eC9 (with PLQY: 2.53% measured in neat film), for which the π -conjugation backbone is extended, resulting in an EL spectrum red-shift to the NIR(II) region of 1050 nm. We then selected the Pt(II) No.2^[49] (see Scheme of Figure S13) as an energy donor, for which the emission is at 800 nm to maximize the overlap with the absorption of BTPV-eC9 (see Figure S13). As a result, for the OLEDs with only BTPV-eC9 as the emitter, the optimized device exhibits an external quantum efficiency (EQE) of 0.083 % and a radiance of 9.69 W sr⁻¹ cm⁻². Upon using the Pt(II) No.2/BTPV-eC9 double layers, we again successfully demonstrate interfacial energy transfer with an optimized architecture ITO/HATCN (10 nm)/NPB (50 nm)/mCP (15 nm)/Pt(II) No.2 (4 nm)/BTPV-eC9/PO-T2T (25 nm)/LiF (1 nm)/Al (120 nm), giving a 1020 nm emission with an EQE of 0.66 % and a radiance of 18.67 W sr⁻¹ cm⁻² (see Figure S14). These results show that the interfacial energy-transfer method is promising for the application of near-infrared OLEDs.”

Finally, we do agree, to certain extent, that we still use the Pt(II) complex as the energy donor. However, these well-known Pt(II) complexes have been produced with a very high yield following the well-known synthetic protocols. Therefore, they are worth pursuing from both synthetic efficiency and cost-effectiveness perspectives.

Figure S13. (a) Chemical structure of Pt(II) No.2 and (b) BTPV-eC9 molecules, along with (c) their energy levels.(d) CVs and DPVs of BTPV-eC9 in DCM/0.1 M TBAP under nitrogen. (e) Absorption and emission spectra of Pt(II) No.2 and BTPV-eC9, with the overlapping region indicating the radiative energy transfer zone.

Figure S14. Radiance and current versus voltage curves: (a) BTPV-eC9 and (d) Pt(II) No.2/BTPV-eC9. EL spectra of (b) BTPV-eC9 and (e) Pt(II) No.2/BTPV-eC9. EQE versus current density curves: (c) BTPV-eC9 and (f) Pt(II) No.2/BTPV-eC9.

3. Line 134: “The decline of BTP-eC9 PLQY upon its spin coating onto Pt(fprpz)₂ layer is attributed to the chemical interaction between BTP-eC9 and Pt(fprpz)₂ layers in the interface”. Authors should clarify what they mean by “chemical interaction”.

Reply: We apologize for the imprecision in our terminology and have therefore replaced "chemical interaction" with "physical contact" that generally refers to van der Waals forces, hydrogen bonds, various molecular stacking, etc. A revised context has been provided in the new manuscript and below.

“The decline of BTP-eC9 PLQY upon its spin coating onto Pt(fprpz)₂ layer is attributed to the physical contacts between BTP-eC9 and Pt(fprpz)₂ layers in the interface. For example, the different evaporation rate of solvents between BTP-eC9 and

Pt(fprpz)₂ layers not only induces an endothermic phenomenon but also leads to the formation of multiple interfaces due to thermal stresses, causing mutual influences on the assembly of molecules.”

4. Line 141: “...stamping technique was rarely applied in OLEDs”. Please provide at least one of such references on transfer printed OLEDs.

Reply: Thank you for the reviewer’s reminding. We have added five corresponding references (numbers 37-41): Nanoscale Horizons 5, 144-149 (2020); Advanced Functional Materials 29, 1902412 (2019); Advanced Materials 8, 245-247 (1996) 245-247.; Nature 403, 750–753 (2000); and ACS Nano, 14, 1133-1140 (2020).

5. Line 150: “The PLQY achieved via the stamping method was 8.85%, higher than the PLQY of BTP-eC9. This enhancement shows that in addition to the energy transfer from Pt(fprpz)₂ to the BTP-eC9 pathway, there are other channels, such as the reabsorption of Pt(fprpz)₂ emission by BTP-eC9”. However, in the Supplementary info file, Table S1, 800 nm excitation wavelength is reported. With this excitation wavelength, no excitation and consequent reabsorption of the Pt(fprpz)₂ emission should occur. In the same Table S1, emission at 754 nm from the complex is also reported, which should not be observed unless energy upconversion is taking place. Authors should clarify this.

Reply: We gratefully acknowledge the reviewer's reminder. Due to a formatting oversight, a typographical error occurred in the original manuscript; we have removed the 754 nm entry from Table S1.

6. Figure 2: There seems be a typo in the legends. λ_{em} should be λ_{exc} , i.e., excitation wavelength.

Reply: Thank you for the reviewer's reminder. We have updated Figure 2 by replacing λ_{em} with λ_{exc} as indicated. Also, please see the updated Figure 2 below for the reviewer’s convenience.

Reviewer #3 (Remarks to the Author):

This paper reports an interesting NIR OLED result employing sensitized emission of a fluorescent molecule by a sensitizing Pt phosphor. The best results are obtained by stamping of one layer onto another forming a bilayer.

Overall, the engineering results are of interest: I am not aware of such a high efficiency from an OLED at such a long wavelength, although it is much inferior to performance of inorganic LEDs whose IQE approach 100%.

There are numerous deficiencies in the paper that need to be pointed out. There are errors in language and terminology as well.

1. The title is hype. Please tone it down. There is too much claimsmanship in papers already so we should be more modest and let the readers decide whether this is a 'leap forward' or not.

Reply: We thank the reviewer for his/her valuable suggestion. We then take the reviewer's suggestion and have revised the title to: "High-Performance Organic Near-Infrared OLEDs through Interfacial Energy Transfer: Peak Wavelength 925 nm, External Quantum Efficiency 2.24%, and Radiant Brightness $39.97 \text{ W sr}^{-1} \text{ m}^{-2}$."

2. Line 96: this is a type I heterojunction, not a quantum well.

Reply: Thank you for the reviewer's suggestion. We have revised it to: "type-I heterojunction-like structure."

3. There are no PL spectra of the mixed or layered structures. This makes it impossible to evaluate the other spectra within the paper.

Reply: Thanks to the reviewer for the reminder, in fact, the steady-state photoluminescence (PL) spectra of the two emissive layers (spin layer and stamp layer) are already included in Figure S1 of the original Supporting Information.

4. There are almost no error bars to the data. The only error noted is in Fig. 5e, but ALL errors to all measured values should be quoted in the abstract and throughout the text. Only then can I determine whether 51.5 ns and 49 ns in Line 128 are significantly different.

Reply: We fully agree with the reviewer and have re-examined the data and added error bars to all necessary available data. The statistical confidence factor value has been added in figure legends of Figure 2 and as following: “Lifetime fits with a statistical confidence factor (χ^2) less than 1.3.” Also, the lifetime values have been measured in four different samples. We then measured the lifetime with an average of four replica. Note that the standard deviation of lifetime depends on the measured time window. For measurement of fast (< 10 ns) and slow ($>> 10$ ns), the time window of 100 ns and 500 ns was commonly applied, which gave an error bar of ± 0.4 and ± 1.2 ns, respectively.

The time-resolved photoluminescence curve of the Pt(fprpz)₂/BTP-eC9 spin-coated film monitored at 750 nm is shown in Figure 2(d). The decay for front and rear excitation was fitted to be $\tau(\text{front}) = 49.0$ and $\tau(\text{rear}) = 51.5$ ns, respectively, which within the experimental uncertainty, are about the same. However, this value is significantly smaller than the emission lifetime (322.0 ns) of the pure vapor deposited Pt(fprpz)₂. Furthermore, in the Pt(fprpz)₂/BTP-eC9 spin-coated film, upon rear-side excitation where Pt(fprpz)₂ was excited directly, the resulted BTP-eC9 920 nm emission was much weaker than that of the front-side excitation where BTP-eC9 was excited (see Figure S1 of Supporting Information). The results clearly indicate lack of energy transfer from Pt(fprpz)₂ to BTP-eC9 in the Pt(fprpz)₂/BTP-eC9 spin-coated film. We thus conclude that the drastic quenching of Pt(fprpz)₂ emission lifetime in the Pt(fprpz)₂/BTP-eC9 spin-coated film does not originate from the energy transfer but from the inferior morphology of the Pt(fprpz)₂ film that is destroyed by mixing with the

BTP-eC9 part upon spin coating. We have added a corresponding statement in the revised text and supporting information. Please see page 5 of the revised text.

5. Discussions of preexponential factors a_1 and a_2 (paragraph starting on line 139) are unclear. Error bars needed. They need to be rigorously defined.

Reply: We highly appreciate reviewer's valuable comments and have made significant revision to improve the clarity. Also, uncertainties in values of pre-exponential values have been added. The relevant statement is also elaborated as follows:

“Due to the efficient energy transfer to BTP-eC9, the emission lifetime of $\text{Pt}(\text{fprpz})_2$ is reduced from 322.0 ns to 100.5 ns, and the energy transfer ratio is calculated to be 68.8%. Therefore, we divide the emission lifetime into two components, where τ_1 (100.5 ns) represents energy transfer, and the other is denoted by τ_2 (322.0 ns) without energy transfer. Their corresponding pre-exponential factor a_1 refers to the proportion of energy transfer and a_2 refers to the proportion without energy transfer. By analyzing a_1 under different conditions, we can obtain information about where the energy transfer occurs. As a result, a_1 of 0.76 ± 0.04 deduced from the front direction is significantly higher than a_1 of 0.11 ± 0.02 detected from the rear side excitation, indicating that most of the energy transfer from $\text{Pt}(\text{fprpz})_2$ occurs close to the BTP-eC9 side, i.e. around the interfacial area.”

6. What is the Förster transfer radius? Is this consistent with the drop in efficiency? Also, the Pt diffusion length is needed for complete analysis of performance. But one thing is apparent: There are significant losses in this device since the Pt efficiency is far greater than the final device efficiency. Where do these losses originate?

Reply: We highly appreciate the reviewer's comments. The calculated Förster transfer radius R_0 is 11 nm. Detailed calculation approach was added in the updated version of Supporting Information. For an average film thickness of about 10 nm, the calculated FRET efficiency is 63.9 %, which is close to the FRET efficiency of 68.9 % obtained from emission lifetime decay. However, the above experiments and calculations are based on photoluminescence. Due to limitations in the sensitivity of the experimental equipment, we had to coat $\text{Pt}(\text{fprpz})_2$ and BTP-eC9 layers much thicker than 11 nm, so

a large part of the $\text{Pt}(\text{fprpz})_2$ emissions remained unchanged. Electroluminescence experiments show that thinner (5 nm) $\text{Pt}(\text{fprpz})_2$ has less donor emission residue. Also, during electrical pumping, we have fine-tuned the carrier mobility through ETL, HTL, and barrier layers, so that electron/hole pairs can recombine near the interface. Therefore, the energy transfer efficiency of the optimized OLED could reach nearly 100% upon optimization.

7. Figures are disorganized. Putting section (c) after (e) is confusing in Fig. 2.

Reply: We apologize for the previously disorganized arrangement of figures and have now reorganized Figure 2. Please also see below for the reviewer's convenience.

8. The GIWAXS analysis is not useful without knowing the crystal structures. There is reference to d-spacings and stacking habits but without the crystal structure this is not a verifiable assertion.

Reply: We thank the reviewer for enlightening this comment. The crux of the matter is that both $\text{Pt}(\text{fprpz})_2$ and BTP-eC9 are highly self-assembled materials that form bulk structures. Therefore, growing single crystal for the x-ray analysis is currently infeasible. Alternatively, probing the film orientation through Grazing Incidence Wide-Angle X-Ray Scattering (GIWAXS) is necessary, which is widely applied by in the field of OLEDs, especially those mixing electron donor and acceptor systems, forming exciplexes in the interface. We agree with the reviewer in that without the crystal structure, one cannot make verifiable assertion only from the reference to d-spacings and stacking habits. Nevertheless, the result should render complementary supports for

the morphology of surface.

9. This is not the first demonstration of stamped combinations of organic layers to form a device (or even an OLED). References to the literature are required.

Reply: Thank you for the reviewer's reminding. We have added five corresponding references (numbers 37-41): *Nanoscale Horizons* 5, 144-149 (2020); *Advanced Functional Materials* 29, 1902412 (2019); *Advanced Materials* 8, 245-247 (1996) 245-247.; *Nature* 403, 750–753 (2000); and *ACS Nano*, 14, 1133-1140 (2020).

10. I don't know what a negative TAS is due to. Simply stating it is due to free charge, chemical contact, etc. is insufficient. Scientific rigor is missing throughout the paper.

Reply: We apologize for the inappropriate interpretation of the data of picosecond-transient absorption spectra (ps-TAS) and, as a result, have reinterpreted the ps-TAS results in the revised manuscript, which is also attached below for the reviewer's convenience.

“In addition to GIWAXS and AFM measurements, we also explored the carrier behaviors possibly influenced by interfaces via picosecond-transient absorption spectra (ps-TAS). ps-TAS has been widely applied to study the carrier extraction behavior of organic photovoltaics (OPVs), yet research on ps-TAS in organic light-emitting diodes (OLEDs) is scarce. BTP-eC9 belongs to the derivatives of Y-family with the same core chromophore as the emitter, so there are a wealth of literature references to provide. In the ps-TAS study of Y-family derivatives, many reports have identified the 780 nm transient absorbance signal as indicative of charge separation (CS) signal^[43-45]. On the other hand, the major transient absorption of the triplet exciton state of BTP-eC9 should be in ~1400 nm region^[43]. Therefore, the observed 780 nm transient positive absorbance can be clearly ascribed to the TAS of the CS signal.

We transfer the concept in OPV but opposite consequence to the NIR-bilayer emitters in OLEDs, where the generation of any CS is undesirable, as it would reduce the charge recombination. Upon 650 nm excitation, in pure BTP-eC9, we did not detect any CS signals, indicating the absence of interfaces (see Figure S6). However, when BTP-eC9 was spin-coated onto Pt(fprpz)₂, a CS signal emerged at 85 ps (Figure 4c), indicating

the creation of certain interfaces due to physical contact such as van der Waal force, hydrogen bonding, π -stacking, etc. in between. Conversely, when BTP-eC9 was transferred onto Pt(fprpz)₂ using a stamping method, the CS signal was delayed to 355 ps (see Figure 4c) with significantly smaller transient absorbance (positive ΔOD , cf. spinning method), suggesting that stamping does not generate a multitude of interfaces. For clarity, Figure 4d schematically illustrates the exciton dynamics around interfaces prepared by different methods. Here, we must emphasize that the ps-TAS measurement is via the optical pumping, which only provides supplementary support for difference of interface formation between spin coating and imprinting techniques. The real influence of interface structure to the OLEDs performance should be probed by electric-pumping, which is unfortunately not feasible due to the much slower time response [46].

»

Figure 4. Pseudo-color plots of the picosecond transient absorption (ps-TA) spectra for (a) Pt(fprpz)₂/BTP-eC9 spin-coated film and (b) Pt(fprpz)₂/BTP-eC9 stamped film. (c) The time-dependent transient absorbance (ΔOD) of Pt(fprpz)₂/BTP-eC9 spin-coated and stamped films monitored at 780 nm. (d) Schematic diagram of exciton dynamics at interfaces prepared by different methods. Process (1): Optically excited exciton generation. Process (2): The spontaneous orientation polarization in the interface

facilitates the charge-transfer (CT) formation and (3) subsequent charge separation (CS). Left of (d): The bulk heterojunction (BHJ) formed due to spin-coating leads to abundant interfaces, providing sufficient driving force for CT formation that undergoes charge separation. Right of (d): The smooth and plane interface produced by stamping facilitates produce less CT states and thus fewer CS separations.

11. There is considerable residual Pt emission. Why? This goes back to my point #6.

Reply: We highly appreciate the reviewer's comments. We calculated the Förster transfer radius R_0 for Pt(fprpz)₂ and BTP-eC9 to be 11 nm (in the revised Supporting Information). In the photoluminescence measurement, due to limitations in the instrumental sensitivity, we had to coat Pt(fprpz)₂ and BTP-eC9 layers much thicker than 11 nm, so a large part of the Pt(fprpz)₂ emissions far away from interface remained unchanged due to lack of interfacial energy transfer. In the EL, the recombination of electron and hole can be harnessed, so EL takes place near the interface between Pt(fprpz)₂ and BTP-eC9. Therefore, the probability of energy transfer is greatly increased because the distance r between Pt(fprpz)₂ and BTP-eC9 is much shorter than the Förster radius R_0 that is calculated to be 11 nm. As a result, much smaller Pt(fprpz)₂ emission was observed. In other words, the energy transfer in near the interfacial zone can be nearly 100%.

In the revised section of simulations, we have also added a more detailed discussion of the residual of Pt(fprpz)₂ emission in electroluminescence. Please see revised text, which is also attached below for the reviewer's convenience.

“By simulating the recombination area distribution for the EL experiment, EL takes place near the interface between Pt(fprpz)₂ and BTP-eC9 (see Supporting Information). Therefore, the probability of energy transfer is greatly increased because the distance r between Pt(fprpz)₂ and BTP-eC9 is much shorter than the Förster radius R_0 calculated to be 11 nm (for detail, see Supporting Information). Note that thicker films will cause some electrons and holes to recombine in areas far from the interface. Especially, when r is $>R_0$ the energy transfer probability is greatly reduced, resulting in non-negligible residual Pt(fprpz)₂ emission.”

12. No analysis is provided to explain the high voltage operation. This needs to be included.

Reply: In devices containing only Pt(fprpz)₂, a turn-on voltage of 4.2 V is required. Consequently, in the cases of bilayer and sandwich devices where the overall film is thicker, it inevitably leads to a delayed turn-on voltage for the device. This phenomenon has been observed in previous reports, for example, J. Phys. Chem. C, 2018, 122(5), 2951–2958.

13. What was the set up to measure OLED EQE? Is it within an integrating sphere, or in the forward direction? What detector was used, how was it calculated?

Reply: We are grateful to the reviewer for his/her valuable comment and suggestion. The methodology employed for measurement is of paramount importance. In response, we have consulted with the instrument supplier (Enlitech) and have supplemented the method section for clarification as follows:

"The LQ-50X system (Enlitech, Taiwan) includes a PTFE integrating sphere, a Multi-Channel Photon Detector (MCD) and two spectrometers to collect emission photons and subsequent spectral analyses. The MCD enhances sensitivity, facilitating effective detection in low-light conditions. The system is capable of measuring a broad wavelength range from 300 to 1700 nm using Si and InGaAs detectors, and it is calibrated against a NIST-traceable standard lamp."

Reviewer #4 (Remarks to the Author):

Hung and co-workers reported a strategy for fabricating NIR-OLEDs relying on an interfacial energy transfer process. Specifically, the authors took advantages of the superior NIR emissions of self-assembled Pt(II) complex and a fluorescence dye (BTP-eC9), which can be maintained using the transfer-printing method. It is interesting to find that the emission of BTP-eC9 can be enhanced in stamped films probably due to changes in its packing arrangement. The authors performed time-resolved photoluminescence, transient absorption and other spectroscopic measurements to characterize the structure and energy transfer process. The device efficiencies were improved by ca. 6-folds in comparison with the fluorescent OLEDs. This work demonstrates a way to advance the efficiency of NIR OLEDs using fluorescent dye as

the terminal emitter. However, there are some issues that need to be addressed before its consideration for publication on Nature Communications.

Reply: We highly appreciate the comments provided by the reviewer. Comments and suggestion are replied item by item as follows.

Major revisions:

1. The interfacial energy transfer mechanism is reminiscent of phosphorescence sensitized fluorescence OLEDs by co-doping the metal complexes and fluorescent emitter. As stated in the manuscript, the authors tried to co-spin coat the mixing Pt(II) complex and BTP-eC9, which was unsuccessful. Therefore, the self-assembled Pt(II) structure is still one of the keys to the achievement in this work, which can effectively suppress exciton-vibrational coupling. From this point of view, it is hard to accept that interfacial energy transfer is really the most or only important factor. I suggest the authors to reorganize the writing with these two considerations: (1) which factor is the most important for NIR-OLEDs using the interfacial energy transfer method? (2) What is the advantage of this interfacial energy transfer method?

Reply: We highly appreciate the comments provided by the reviewer and have attempted to reply, to our best, the above two specific suggestions in the introduction section, which is also elaborated as follows for the reviewer's convenience.

“Although the application of transfer printing in OLED is rare, there are also sporadic reports. However, most reports focus on the transporting layer^[26] rather than the emission layer, which is the core of this study. To gain 100% internal conversion efficiency for the regular NIR fluorescence dye, the exploitation of energy transfer from either triplet or TADF donors is inevitable. As for the emission near NIR(II) region (~1000 nm), up to this stage, only those self-assembled Pt(II) complexes can fulfill the gap for energy transfer. However, the complicated NIR fluorescence dye prohibits the vapor deposition. Alternatively, exploiting the mixing method, the self-assembled structure of the Pt(II) complexes may be destroyed. As a result, the interfacial energy transfer provides a unique solution for the generation of NIR-OLEDs.”

2. Following Comment 1, in Line 56, the authors pointed out the limitation of using precious Pt element for NIR OLEDs based on Pt(II) aggregates with triplet MMLCT excited states. This is true. But, for the present interfacial energy transfer strategy, Pt(II)

is still required and key to the device performance. So, it might be not acceptable to attract readers by using “Fluorescent Near-infrared OLEDs”. I suggest the authors to revise the writing in this aspect.

Reply: We thank the reviewer for his/her valuable suggestion. We then take the reviewer’s suggestion and have toned down the “fluorescent near-infrared OLEDs” relevant statement by “hyperfluorescent near-infrared OLED”, which is generated by triplet-to-singlet energy transfer, throughout the text and Supporting Information.

3. In Lines 72-74, the authors remarked the QD-LEDs and PeLEDs with EQEs >25. There have many reports on OLEDs having high EQEs even close to 40%. Preferred horizontal transition dipole moment has been identified as a key parameter dictating the device EQEs. Because the present work is on NIR OLEDs, references should not be only limited to QD-LEDs and PdLEDs. On the other hand, are the self-assembled layers beneficial for increasing the light out-coupling efficiency?

Reply: We concur with the reviewer's comments and have accordingly added two references (numbers 24 and 25 from Adv. Mater., 33, 2008032 (2021) and ACS Materials Lett., 5, 2339-2347 (2023), respectively) regarding the outcoupling effect for OLEDs. Following the reviewer’s suggestion, we conducted measurements of light out-coupling, and have included the results and corresponding discussion (see below) in the revised manuscript.

"We also measured the horizontal dipole ratios, where the Θ value for Pt(fprpz)₂ alone was found to be 78%, indicating the effect of light out-coupling. In contrast, the Θ value for BTP-eC9 was 67%, and for the Pt(fprpz)₂/BTP-eC9 (stamp) monitored at 950 nm, the Θ value remained at 67%, demonstrating that interface energy transfer does not affect the light out-coupling efficiency. (See Figure S11)"

Figure S11. Horizontal dipole ratios (a) Pt(fprpz)₂ (b) BTP-eC9 (c) Pt(fprpz)₂/BTP-eC9

(stamp)

4. The energy transfer from Pt(fprpz)₂ to BTP-eC9 is described as efficient throughout the text. But there are significant portion of emission from the Pt(II) donor in the photoluminescence test. In host-guest system, the efficient energy transfer is usually characterized by a complete attenuation of the donor emission. Although the lifetime is reduced from 322 ns to 100.5 ns, this may not be used as an indicator for this judgement. Also, the device shows emissions from both the Pt(II) and BTP-eC9, revealing that the tuning of energy transfer is not very successful. Although the authors tried to optimize the device structures according to simulation results, they have not tackled this problem. This is very important for evaluating this interfacial energy transfer concept for the design of NIR OLEDs.

Reply: We highly appreciate the reviewer's comments. We calculated the Förster transfer radius R_0 for Pt(fprpz)₂ and BTP-eC9 to be 11 nm (in the revised Supporting Information). In the photoluminescence measurement, due to limitations in the instrumental sensitivity, we had to coat Pt(fprpz)₂ and BTP-eC9 layers much thicker than 11 nm, so a large part of the Pt(fprpz)₂ emissions far away from interface remained unchanged due to lack of interfacial energy transfer. In the EL, the recombination of electron and hole can be harnessed so EL takes place near the interface between Pt(fprpz)₂ and BTP-eC9. Therefore, the probability of energy transfer is greatly increased because the distance r between Pt(fprpz)₂ and BTP-eC9 is much shorter than the Förster radius R_0 (11 nm). As a result, much weaker and even negligible Pt(fprpz)₂ emission (quenched by energy transfer) was observed. In other words, the energy transfer in near the interfacial zone can be nearly 100%.

In the revised section of simulations, we have also added a more detailed discussion of the residual of Pt(fprpz)₂ emission in electroluminescence. Please see revised text and paragraph attached below for the reviewer's convenience.

“By simulating the recombination area distribution for the EL experiment, EL takes place near the interface between Pt(fprpz)₂ and BTP-eC9 (see Supporting Information). Therefore, the probability of energy transfer is greatly increased because the distance r between Pt(fprpz)₂ and BTP-eC9 is much shorter than the Förster radius R_0 calculated to be 11 nm (for detail, see Supporting Information). Note that thicker films will cause

some electrons and holes to recombine in areas far from the interface. Especially, when r is $>R_0$ the energy transfer probability is greatly reduced, resulting in non-negligible residual Pt(fprpz)₂ emission.”

5. The authors should elaborate more on the formation of the exciton dissociated (ED) states at the interfaces. For example, how would the band alignment affect this process?

Reply: We apologize for the inappropriate interpretation of the data of picosecond-transient absorption spectra (ps-TAS) and, as a result, have reinterpreted the ps-TAS results in the revised manuscript, which is also attached below for the reviewer’s convenience.

“In addition to GIWAXS and AFM measurements, we also explored the carrier behaviors possibly influenced by interfaces via picosecond-transient absorption spectra (ps-TAS). ps-TAS has been widely applied to study the carrier extraction behavior of organic photovoltaics (OPVs), yet research on ps-TAS in organic light-emitting diodes (OLEDs) is scarce. BTP-eC9 belongs to the derivatives of Y-family with the same core chromophore as the emitter, so there are a wealth of literature references to provide. In the ps-TAS study of Y-family derivatives, many reports have identified the 780 nm transient absorbance signal as indicative of charge separation (CS) signal [43-45]. On the other hand, the major transient absorption of the triplet exciton state of BTP-eC9 should be in ~1400 nm region [43]. Therefore, the observed 780 nm transient positive absorbance can be clearly ascribed to the TAS of the CS signal.

We transfer the concept in OPV but opposite consequence to the NIR-bilayer emitters in OLEDs, where the generation of any CS is undesirable, as it would reduce the charge recombination. Upon 650 nm excitation, in pure BTP-eC9, we did not detect any CS signals, indicating the absence of interfaces (see Figure S6). However, when BTP-eC9 was spin-coated onto Pt(fprpz)₂, a CS signal emerged at 85 ps (Figure 4c), indicating the creation of certain interfaces due to physical contact such as van der Waal force, hydrogen bonding, π -stacking, etc. in between. Conversely, when BTP-eC9 was transferred onto Pt(fprpz)₂ using a stamping method, the CS signal was delayed to 355 ps (see Figure 4c) with significantly smaller transient absorbance (positive ΔOD , cf. spinning method), suggesting that stamping does not generate a multitude of interfaces.

For clarity, Figure 4d schematically illustrates the exciton dynamics around interfaces prepared by different methods. Here, we must emphasize that the ps-TAS measurement is via the optical pumping, which only provides supplementary support for difference of interface formation between spin coating and imprinting techniques. The real influence of interface structure to the OLEDs performance should be probed by electric-pumping, which is unfortunately not feasible due to the much slower time response [46].

”

Figure 4. Pseudo-color plots of the picosecond transient absorption (ps-TA) spectra for (a) Pt(fprpz)₂/BTP-eC9 spin-coated film and (b) Pt(fprpz)₂/BTP-eC9 stamped film. (c) The time-dependent transient absorbance (ΔOD) of Pt(fprpz)₂/BTP-eC9 spin-coated and stamped films monitored at 780 nm. (d) Schematic diagram of exciton dynamics at interfaces prepared by different methods. Process (1): Optically excited exciton generation. Process (2): The spontaneous orientation polarization in the interface facilitates the charge-transfer (CT) formation and (3) subsequent charge separation (CS). Left of (d): The bulk heterojunction (BHJ) formed due to spin-coating leads to abundant interfaces, providing sufficient driving force for CT formation that undergoes charge separation. Right of (d): The smooth and plane interface produced by stamping facilitates produce less CT states and thus fewer CS separations.

6. Discussion on the charge transport and recombination in devices is missing. With two or three emission layers, the recombination zones should be crucial to the electroluminescence spectra and efficiencies.

Reply: Thank you for the reviewer's valuable feedback. We have added measurements and descriptions of the charge mobility in the two emission layers to the manuscript, which is also elaborated below.

"Prior to the fabrication of NIR-OLEDs, it is essential to gain insight into the charge transport and recombination dynamics. Hence, we analyzed the charge mobilities of the emission layers using the Space charge-limited current (SCLC) technique. For the hole-only device configuration (ITO/MoO₃/Emission layers/MoO₃/Al), the hole mobility of Pt(fprpz)₂ was measured at $3.95 \times 10^{-6} \text{ cm}^2 \text{ V}^{-1} \text{ s}^{-1}$, and for BTP-eC9, it was $4.22 \times 10^{-5} \text{ cm}^2 \text{ V}^{-1} \text{ s}^{-1}$ (see Figure S7). In the electron-only device configuration (ITO/Al/Emission layers/Al), the electron mobility for Pt(fprpz)₂ was found to be $2.67 \times 10^{-5} \text{ cm}^2 \text{ V}^{-1} \text{ s}^{-1}$, and for BTP-eC9, it was $1.59 \times 10^{-5} \text{ cm}^2 \text{ V}^{-1} \text{ s}^{-1}$ (see Figure S7). Based on the SCLC results, we can infer that the majority of emission occurs in BTP-eC9. This is attributed to the matched electron mobility, allowing for efficient electron transfer to BTP-eC9. Although there is a significant difference in hole mobility, the lower HOMO level of Pt(fprpz)₂ by 0.4 eV compared to BTP-eC9 ensures that holes still migrate towards BTP-eC9 under an applied bias."

Figure S7. SCLC fitting for the $J^{0.5}$ - V curve of hole-only and electron-only devices.

Minor revisions:

1. Transfer printing in OLEDs has not been widely explored. I suggest the authors to add key results/progresses on this topic in the Introduction.

Reply: We fully appreciate the reviewer's comment and suggestion. Accordingly, we have added some crucial references on transfer-printed OLEDs and transfer-printed devices (see references 37-41). However, the cited references were all focused on the transporting layer, not the emissive layer that is the core of this study. For clarity, a relevant statement has been added in the introduction section, which is also written below.

“Note that the applications of transfer printing in OLEDs, though being rare, have been reported sporadically. However, most of reports were focused on the transporting layer, not the emissive layer that is the core of this research [26].”

2. Please indicate the methods for determining the MO levels for both Pt(II) and BTP-eC9, including their states in either solution or film.

Reply: Thank you for raising this query. The quantum computation for both Pt(fprpz)₂ and BTP-eC9 have been previously published. The HOMO and LUMO levels of Pt(fprpz)₂ were taken from our earlier publication in Nat. Photonics 11, 63-68 (2017), while the properties of BTP-eC9 were referred to Adv. Mater. 32, 1908205 (2020).

3. In Line 143: “.....the energy transfer rate is calculated to be 68.8%.” I think this data means energy transfer ratio.

Reply: Sorry for the mistake. It has been corrected.

REVIEWER COMMENTS

Reviewer #1 (Remarks to the Author):

The authors have followed up on my comments about generality and mechanism via exciton dissociation/charge separated state.

- They address generality by introducing a second functional emitter (BTPVec9) where the extension to infrared emitting devices is impressive. However it is not clear how others can follow the authors in future. I don't think this has been fully addressed. What are the design rules?

- The authors have reinterpreted their optical spectroscopy data and assign 780 nm to a charge separated signal. They rule out the triplet exciton from literature references of other materials in the same Y-family of electron acceptors for organic photovoltaics, which is reasonable. However the 780 nm could still be attributed to a charge transfer state signal bound to the interface? The discussion of the photophysics is confusing and difficult to follow. The authors should provide a Jablonski diagram and perform some modelling of gibbs free energy for the exciton, charge transfer and charge separated states to verify their model. I believe the discussion of the mechanism is still speculative and not currently supported without this analysis or other quantification.

I think the manuscript should only be considered for publication once the authors take more significant steps to address my comments. The device results are impressive. My comments aim to pull out mechanistic understanding that would bring the manuscript to the level for applicability to the readership of Nature Communications.

I will not follow up on comments raised by other reviewers, but very good points were made and should also be addressed.

Reviewer #2 (Remarks to the Author):

Some revisions are still needed, as detailed in the comments below:

1. (regarding Comment 2 in my previous revision) In the introduction, line 54 of the updated manuscript, authors point out that “Despite the above success, the unique features for the Pt(II)

complex may impose the limitation in both breadth and versatility of applications due to the required precious Pt(II) element and specific ligands.” I agree with the authors’ point for using Pt-complexes, but this sentence should be removed or amended as it sounds misleading. Authors still use Pt-complexes in the devices presented in this manuscript.

2. (former comment 3, about what authors meant by “chemical contact”) Switching from chemical to physical contact and vaguely referring to “van der Waals forces, hydrogen bonds, various molecular stacking...” is not sufficient. Also, I cannot find the revised text quoted in the rebuttal letter referring to an endothermic process in the updated manuscript.

3. (former comment 5, about reabsorption of the Pt(fprpz)₂ emission). It is still not clear how you can get reabsorption of the emission from the Pt-complex when the bilayer Pt(fprpz)₂/BTP-eC9 is excited at 800 nm, at which the complex does not absorb and therefore cannot emit. I would expect the table to include the PLQY obtained by exciting the bilayer at lower wavelengths (e.g., 505 nm). Please clarify.

4. (former comment 6). There is still λ_{em} which should be λ_{exc} in panel b of figure 2.

Reviewer #3 (Remarks to the Author):

I appreciate the authors' response to the first set of reviews. While many questions have been answered, many remain that still need attention before this work can be considered ready for publication.

1. I mentioned the lack of error bars. This problem persists in the revised manuscript. Every experimental number must have errors reported, especially in the abstract, introduction and conclusions as well as in the body of the text. Without knowing the errors, it is impossible to evaluate the importance of a particular result. This is standard practice in scientific papers and I am surprised that it is missing here.

2. In discussing Eq. 1 the statement is made: "N is the exciton delocalization length. which, in theory, can be reduced

by orderly molecular assembly that increases exciton delocalization". Unless I misunderstand the point, I believe this is incorrect. N is increased due to increased delocalization, which is increased by order. This, in turn, decreases the reorganization energy which is going in the wrong direction for reducing energy gap losses. So the logic appears incorrect and inverted for getting small energy gap devices to have high efficiency.

3. Incorrect logic appears again with the statement on line 112 "its LUMO energy level of -3.87 eV closely aligns with that

of Pt(fprpz)₂, impeding energy transfer". The LUMO has nothing to do with energy transfer, only the exciton energies.

4. I am puzzled by the very fast decay of the Pt phosphor. At the very high radiative rate of 322 ns, one would expect strong non-radiative decay, yet the claim is that it has 80% PLQY. This is the fastest phosphor by far in this case. Given that the transition is via MMLCT, one would expect radiative decay times in the μs time scale. However, even going to ref. 31 it is unclear what is going on here. This confusion needs to be cleared up. This short lifetime appears again on line 155.

5. Line 138 "Note that the standard deviation of lifetime under an average of four replica gave an error bar of ± 1.2 ns." What does this mean?

6. Lines 222 -228: The statement "Based on the SCLC

results, we can infer that the majority of emission occurs in BTP-eC9. This is attributed to the matched electron mobility, allowing for efficient electron transfer to BTP-eC9. Although there is a significant difference in hole mobility, the lower HOMO level of Pt(fprpz)₂ by 0.4 eV compared to BTP-eC9 ensures

that holes still migrate towards BTP-eC9 under an applied bias." is very confusing. I had thought that the mechanism for exciting the BTP-eC9 was exciton transfer from the Pt complex. Yet this statement clearly states that the emission is attributed to electron transfer to the fluorophore. Which is it? Electron or exciton transfer? It cannot be both. I am afraid the authors are actually unclear as to what is happening in their device.

7. Line 252: Here the losses are clearly significant. Only a fraction of the excitations on Pt end up on BTP-eC9. Nonradiative decay seems large once again, contradicting the 80% PLQY and fast time constant for triplet decay on the Pt complex.

There seems to be significant confusion as to mechanisms concluded in this paper that need clarification before this is acceptable.

Reviewer #4 (Remarks to the Author):

The authors have addressed my concerns in the revised manuscript. It is now suitable for publication.

Reviewer #1 (Remarks to the Author):

The authors have followed up on my comments about generality and mechanism via exciton dissociation/charge separated state.

- They address generality by introducing a second functional emitter (BTPVec9) where the extension to infrared emitting devices is impressive. However it is not clear how others can follow the authors in future. I don't think this has been fully addressed. What are the design rules?

We are grateful for the reviewer's valuable input. In fact, we are currently experimenting with using Perovskite LED (FAPbI₃) or Quantum Dot LED (CdSe) as energy donors, paired with fluorescence NIR dyes that have corresponding absorption spectra as energy acceptors. The criteria we have formulated have been added to the manuscript (see also the attachment below).

“Ultimately, we have deduced that the design of an interfacial energy transfer device must satisfy the following criteria: 1) The photoluminescence of the energy donor must overlap with the absorption spectrum of the energy acceptor; the former requires a strong photoluminescence quantum yield (PLQY) and the latter necessitates a high absorption coefficient. 2) For an energy donor and an energy acceptor to function effectively, there must be a sufficient difference between their energy levels. If the LUMO levels overlap, this would lead to the charge distributed evenly, rather than localized at the interface, resulting in an adverse effect. Similarly, the HOMO levels would exhibit analogous phenomena. 3) In OLEDs, the device must be optimized to bring the electron-hole recombination near the interfacial zone within the effective Förster resonance energy transfer (FRET) distance, so that energy can be transmitted.” Upon meeting these criteria, we believe that the stamp interface energy transfer method could expedite the improvement of NIR OLEDs efficiency.

Here, we have to emphasize it again. Since FRET efficiency is inversely proportional to the sixth power of the donor-acceptor distance, when utilizing the FRET effect to produce luminescent OLEDs, donors and acceptors are usually mixed together to reduce the donor-acceptor distance, thereby enhancing the FRET effect. Unfortunately, mixing is not suitable for certain materials that require either an ordered arrangement or readily self-assembling such as BTP-eC9 and Pt(fprpz)₂, respectively, to boost the emission. Alternatively, this study incorporates computer simulation to optimize the type and thickness of HTL, ETL and the relative thickness of the double layer, and then conducts experimental fine-tuning for OLEDs. Ultimately, we would be able to locate the recombination of electron and hole near the interface and allows efficient Förster-type resonance (i.e., dipole resonance) energy transfer to occur.

In sum, for materials that require an ordered arrangement of both donors and acceptors to achieve good quantum efficiency and longer emitting wavelength in NIR region, the

bi-layer stamp coating method can effectively maintain the ordering of the two layers. Coupled with fine-adjusting the recombination region of electrons and holes, this methodology can greatly improve the efficiency of FRET and improve the performance of the final device. The current study thus serves as a paradigm of the interfacial FRET, especially for those emitters requiring ordered packing to enhance the OLED performance.

- The authors have reinterpreted their optical spectroscopy data and assign 780 nm to a charge separated signal. They rule out the triplet exciton from literature references of other materials in the same Y-family of electron acceptors for organic photovoltaics, which is reasonable. However the 780 nm could still be attributed to a charge transfer state signal bound to the interface? The discussion of the photophysics is confusing and difficult to follow. The authors should provide a Jablonski diagram and perform some modelling of Gibbs free energy for the exciton, charge transfer and charge separated states to verify their model. I believe the discussion of the mechanism is still speculative and not currently supported without this analysis or other quantification.

Reply: We are grateful to the reviewer for his/her valuable comments. Following the reviewers' suggestion, we have supplemented our manuscript with a Jablonski diagram with detailed description provided in the revised text. For the convenience of the reviewer, please also see below

Figure 3. A Jablonski diagram to demonstrate an overview of the role of interfacial energy transfer dynamic processes underlying NIR OLED functionality. The solid sky-blue pathway represents the process of interfacial energy transfer, capable of facilitating FRET. Within this framework, the S₀ and S₁ states denote the ground and excited states, respectively, in the singlet manifold, while the T₁ state represents the triplet state. Note that T₁ → S₁' FRET is viable because the T₁ → S₀ transition is virtually allowed for the Pt (II) complex due to its strong spin-orbit coupling. The diagram also outlines alternative pathways with dashed lines, where charge transfer (CT) and charge transfer-triplet (CT-T) states are included,

alongside the charge separation (CS) state, which represent subsidiary processes occurring with rather small probability.

We then added a paragraph in the revised text to elaborate content of Figure 3 (see page 7), which is also attached below for the reviewer's convenience.

“The depiction of interfacial energy transfer mechanisms is illustrated in **Figure 3**, where the pathway marked in sky blue lines represents the primary process for the interfacial energy transfer. Regarding Pt(fprpz)₂, its involvement in metal-to-ligand charge transfer (MLCT) and metal-to-metal-to-ligand charge transfer (MMLCT), facilitated by heavy Pt atom and Pt-Pt interaction, respectively, induces an ultrafast intersystem crossing (ISC) with a rate constant of $k_{isc} > 10^{12} \text{ s}^{-1}$, thereby populating the T₁ state with ~100% efficiency. The strong spin-orbit coupling and hence mixing with the singlet-manifold leads the T₁ → S₀ transition virtually allowed, which is evidenced by its sub-microsecond radiative lifetime of phosphorescence^[42]. Conversely, the T₁' state of BTP-eC9, being spin-forbidden for the T₁' → S₀' transition, necessitates that the energy transfer from the T₁ state of Pt(fprpz)₂ to the T₁' state of BTP-eC9 employs an electron exchange mechanism, namely the Dexter-type energy transfer. This electron-exchange type energy transfer requires overlap of electronic wavefunction, and hence takes place within a short distance (e.g., < 1.5 nm), which should be rather inefficient. Additionally, at the interface, loosely bound singlet (triplet) excitons may form, which subsequently undergo dissociation to form free carriers^[43,44]. This process may have small branching ratio but cannot be completely neglected, which will be elucidated in the section of picosecond transient absorption measurement.”

I think the manuscript should only be considered for publication once the authors take more significant steps to address my comments. The device results are impressive. My comments aim to pull out mechanistic understanding that would bring the manuscript to the level for applicability to the readership of Nature Communications.

I will not follow up on comments raised by other reviewers, but very good points were made and should also be addressed.

Reviewer #2 (Remarks to the Author):

Some revisions are still needed, as detailed in the comments below:

1. (regarding Comment 2 in my previous revision) In the introduction, line 54 of the updated manuscript, authors point out that “Despite the above success, the unique features for the Pt(II) complex may impose the limitation in both breadth and versatility of applications due to the required precious Pt(II) element and specific ligands.”. I agree

with the authors' point for using Pt-complexes, but this sentence should be removed or amended as it sounds misleading. Authors still use Pt-complexes in the devices presented in this manuscript.

Reply: We agree with the reviewer's suggestion and have removed the corresponding sentence from the text.

2. (former comment 3, about what authors meant by "chemical contact") Switching from chemical to physical contact and vaguely referring to "van der Waals forces, hydrogen bonds, various molecular stacking..." is not sufficient. Also, I cannot find the revised text quoted in the rebuttal letter referring to an endothermic process in the updated manuscript.

Reply: We apologize for our inadvertence in not incorporating the content set out in the previous reply into the revised manuscript. In response to the reviewer's concerns, we have provided the video and added a corresponding statement to the revised text, which is also attached below. We hope these changes will convince the reviewers.

"----- the decrease in PLQY of BTP-eC9 after spin-coating onto the Pt(fprpz)₂ layer can be attributed to the physical contact between BTP-eC9, solvent, and Pt(fprpz)₂ layer at the interface. The different evaporation rates of the solvent between the two layers of BTP-eC9 and Pt(fprpz)₂ will generate thermal stress, leading to the formation of multiple interfaces and seriously damaging the assembly of the molecules. Alternatively, we employ a stamping technique to transfer solid-state BTP-eC9 directly onto the Pt(fprpz)₂ layer, thus avoiding unnecessary adverse interfacial interactions (see Figure S2 and videos of both methods provided in supporting information)."

In the video (supporting video), it can be observed that the initial film is very glossy and smooth. However, after the spin-coating process, the overall film becomes dull and uneven, and the thickness of the film is evidently uneven and not as expected. The result indicates that the physical contacts by the spin-coating process affect the stacking of Pt(fprpz)₂. On the other hand, through the stamping process (supporting video), it can be seen that the overall film remains glossy and smooth, indicating that the stamping process simply transfers BTP-eC9 onto Pt(fprpz)₂ without affecting the stacking of Pt(fprpz)₂ (see also Figure S2 below).

Figure S2. Films after different post-processing processes, from left to right: original, spin-processed and stamped with BTP-eC9. (a) Well-aligned $\text{Pt}(\text{fprpz})_2$ layer. (b) After the chloroform solvent rinse and spin coating process, it is destroyed into an amorphous $\text{Pt}(\text{fprpz})_2$ film. (c) Amorphous $\text{Pt}(\text{fprpz})_2/\text{BTP-eC9}$ bilayer affected by spin-coated BTP-eC9. (d) Well-aligned $\text{Pt}(\text{fprpz})_2/\text{BTP-eC9}$ bilayer through the imprinting process.

3. (former comment 5, about reabsorption of the $\text{Pt}(\text{fprpz})_2$ emission). It is still not clear how you can get reabsorption of the emission from the Pt-complex when the bilayer $\text{Pt}(\text{fprpz})_2/\text{BTP-eC9}$ is excited at 800 nm, at which the complex does not absorb and therefore cannot emit. I would expect the table to include the PLQY obtained by exciting the bilayer at lower wavelengths (e.g., 505 nm). Please clarify.

Reply: When the bilayer $\text{Pt}(\text{fprpz})_2/\text{BTP-eC9}$ is excited at 800 nm, the Pt complex does not absorb the photon and thus cannot be excited. Therefore, it is equivalent to the excitation of only BTP-eC9. However, PLQY increases from 5.57% ($5.42 \pm 0.17\%$) for only BTP-eC9 on the glass to 8.85% ($8.61 \pm 0.21\%$) for bilayer $\text{Pt}(\text{fprpz})_2/\text{BTP-eC9}$. The improvement in quantum efficiency is due to more organized BTP-eC9 stamped

on the assembled Pt(fprpz)₂. Support of this viewpoint is given by the GIWXAS measurement (see Figure S5), where the reduction of d-spacing of the parallel planes of BTP-eC9 was observed upon stamping on Pt(fprpz)₂.

Following the reviewer's suggestion, we have added each PLQY of dual emission obtained by exciting the bilayer at 505 nm in revised Table S1 (see also below). To avoid confusion, we additionally marked the wavelength range in which each PLQY was analyzed, where 650-850 nm is ascribed to the emission of Pt(fprpz)₂, and 850-1200 nm belongs to the emission of BTP-eC9.)

Table S1. The emission wavelength, photoluminescence lifetime, and photoluminescence quantum yield (PLQY) of different types of thin films.

Sample	Excited (nm)	λ_{em}^{max} (nm)	QY (%) of champion (average \pm error)	Sources for obtaining PLQY ^a
BTP-eC9	800	940	5.57 (5.42 \pm 0.17)	BTP-eC9
Pt(fprpz) ₂	505	750	78.6 (77.6 \pm 0.92)	Pt(fprpz) ₂
Pt(fprpz) ₂ / spin solvent destruction	505	735	0.92 (0.64 \pm 0.24)	Pt(fprpz) ₂
Pt(fprpz) ₂ /BTP-eC9 spin	800	928	5.04 (5.00 \pm 0.05)	Pt(fprpz) ₂
Pt(fprpz) ₂ /BTP-eC9 stamp	800	925	8.85 (8.61 \pm 0.21)	BTP-eC9
Pt(fprpz) ₂ /BTP-eC9 spin	505	735	0.62 (0.51 \pm 0.09)	Pt(fprpz) ₂
		928	4.65 (4.53 \pm 0.12)	BTP-eC9
Pt(fprpz) ₂ /BTP-eC9 stamp	505	750	7.95 (7.79 \pm 0.14)	Pt(fprpz) ₂
		925	7.12 (6.96 \pm 0.15)	BTP-eC9

^a 650-850 nm belongs to the emission of Pt(fprpz)₂, and 850-1200 nm belongs to the emission of BTP-eC9.

4. (former comment 6). There is still λ_{em} which should be λ_{exc} in panel b of figure 2.

Reply: Figure 2b has been corrected. We do appreciate the reviewer's careful examination.

Reviewer #3 (Remarks to the Author):

I appreciate the authors' response to the first set of reviews. While many questions have been answered, many remain that still need attention before this work can be considered ready for publication.

1. I mentioned the lack of error bars. This problem persists in the revised manuscript.

Every experimental number must have errors reported, especially in the abstract, introduction and conclusions as well as in the body of the text. Without knowing the errors, it is impossible to evaluate the importance of a particular result. This is standard practice in scientific papers and I am surprised that it is missing here.

Reply: We fully agree with reviewer's opinion. In the previous version, aimed at showing the best record as well as to avoid reading complexity, we kept a single value without errors in certain places and displayed the average error of the overall data. In this second revised version, we have added PLQY, PL lifetime, and EQE error bars, separately.

2. In discussing Eq. 1 the statement is made: "N is the exciton delocalization length, which, in theory, can be reduced by orderly molecular assembly that increases exciton delocalization". Unless I misunderstand the point, I believe this is incorrect. N is increased due to increased delocalization, which is increased by order. This, in turn, decreases the reorganization energy which is going in the wrong direction for reducing energy gap losses. So the logic appears incorrect and inverted for getting small energy gap devices to have high efficiency.

Reply: We apologize for any misunderstandings caused by our grammatical errors in the original sentence: "where λ_{eff} is the effective reorganization energy of the aggregate states, λ_{M} is the reorganization energy of these promoting modes, and N is the exciton delocalization length. **Which, in theory,** can be reduced by orderly molecular assembly that increases exciton delocalization."

The above statement has been revised to:

"where λ_{eff} is the effective reorganization energy of the aggregate states, λ_{M} is the reorganization energy of these promoting modes, and N is the exciton delocalization length. **In theory,** λ_{eff} can be reduced by orderly molecular assembly that increases exciton delocalization."

3. Incorrect logic appears again with the statement on line 112 "its LUMO energy level of -3.87 eV closely aligns with that of Pt(fprpz)₂, impeding energy transfer". The LUMO has nothing to do with energy transfer, only the exciton energies.

Reply: We are sorry that due to the omission of several intermediate reasoning processes, the reviewer believes that perhaps there are logical errors in our statement. Yes, we fully agree with the reviewer in that LUMO has nothing to do with energy transfer, only the exciton energies for FRET. However, the subject of concern here is the application of OLEDs and we must consider other potentially interfering processes that compete with the energy transfer. Therefore, we performed the computer simulation, followed by the optimization of the OLEDs. In the computation simulation, the LUMO energy level of -3.87 eV for Y11 is close to the LUMO energy level (-3.85 eV) of Pt(fprpz)₂, which makes possible the migration of electrons from Y11 to

Pt(fprpz)₂. This may lead to a recombination not in the side of Y11/Pt(fprpz)₂, resulting in the suppression of energy transfer. In yet another approach, as shown in the Figure S16 (see also below), when we changed LUMO energy from -4.1 eV (BTP-eC9) to -3.9 eV (Y11) that is even closer to LUMO of Pt(fprpz)₂, according to the simulation, the recombination region near the hole transport layer is obviously towards the mCP/Pt-complex interface (see Figure S17 and below), which is expected to give less efficiency in energy transfer. Experimentally, we also fabricated devices incorporating Y11 dye and hence a combination of Pt(fprpz)₂/Y11. This gave an inferior device performance (see Figure S16 and Table S3), supporting the predictions made by Setfos simulations.

Figure S17. The simulation of electron-hole recombination zone using Setfos software. Recombination distribution in devices containing different LUMO levels of energy acceptor, -3.9 eV for Y11 and -4.1 eV for BTP-eC9.

We have accordingly modified the relevant sections in the revised manuscript, which are also attached below for the convenience of the reviewer.

Page 3

“While another NIR dye Y11, as previously reported by Xie et al.^[18], displays a better PLQY, enhanced EQE and stronger emission radiance than that of pure BTP-eC9, in the current study exploiting interfacial energy transfer, Y11 did not have better

performance than that of BTP-eC9 in the sandwich OLEDs. This will be discussed later in the section of device performance.”

Page 11

" The NIR dye Y11 is also studied here, with single-layer dye devices showing enhanced EQE and stronger emission radiance in comparison to that of BTP-eC9. Despite these advantages, however, the LUMO energy level of -3.87 eV for Y11 is nearly in line with that of -3.85 eV for Pt(fprpz)₂, resulting in incomplete charge dispersion at the interface. Figure S16 presents the performance of the Pt(fprpz)₂/Y11/Pt(fprpz)₂ device. Although Pt(fprpz)₂/Y11/Pt(fprpz)₂ exhibits interfacial energy transfer similar to that of Pt(fprpz)₂/BTP-eC9/Pt(fprpz)₂, its device performance is slightly deficient compared to that of Pt(fprpz)₂/BTP-eC9/Pt(fprpz)₂. We have compiled pertinent data in Table S3. We therefore believe that overly similar energy levels may affect the distribution of charges and thus the performance of NIR OLEDs."

Figure S16. Radiance and current versus voltage curves: (a) Y11 and (d) Pt(fprpz)₂ (5nm)/Y11 (stamp)/ Pt(fprpz)₂ (5nm). EL spectra of (b) Y11 (see inset for the structure of Y11) and (e) Pt(fprpz)₂ (5nm)/Y11 (stamp)/ Pt(fprpz)₂ (5nm). EQE versus current density curves with device energy level diagram: (c) Y11 and (f) Pt(fprpz)₂ (5nm)/Y11 (stamp)/ Pt(fprpz)₂ (5nm).

Table S3: EL performance of Y11, BTP-eC9 and BTPV-eC9 NIR OLEDs.

Emitter	V _{on} (V)	R/J/V (W sr ⁻¹ m ⁻² /mA cm ⁻² /V)	EQE _{max} (%) (average ± error)	λ _{max} (nm)
Y11 ^a	1.5	25.88/1915/5	0.32 (0.23±0.07)	954

Sandwiched (Y11) ^c	7.0	33.41/983/15	1.22 (1.08±0.15)	662 (8.6%), 926 (91.4%)
BTP-eC9 ^a	1.3	18.81/2686/4.8	0.18 (0.14±0.04)	952
Sandwiched (BTP-eC9) ^c	6.2	39.97/414/17	2.24 (1.94±0.18)	682 (3.9%), 925 (96.1%)
BTPV-eC9 ^b	1.1	9.69/2946/7	0.08 (0.06±0.02)	1052
Bilayer (BTPV-eC9) ^d	7.4	18.67/573/16	0.66 (0.55±0.10)	778 (3.5%), 1022 (96.5%)

The device structure of ^a is ITO/PEDOT:PSS/PVK/Emitter/C60/BCP/Ag, ^b is ITO/PEDOT:PSS/Emitter/PFN-Br/Ag, ^c is ITO/HATCN/NPB/mCP/Pt(fprpz)₂/Emitter/Pt(fprpz)₂/PO-T2T/LiF/Al and ^d is ITO/HATCN/NPB/mCP/Pt(II) No. 2/Emitter /PO-T2T/LiF/Al.

4. I am puzzled by the very fast decay of the Pt phosphor. At the very high radiative rate of 322 ns, one would expect strong non-radiative decay, yet the claim is that it has 80% PLQY. This is the fastest phosphor by far in this case. Given that the transition is via MMLCT, one would expect radiative decay times in the μ s time scale. However, even going to ref. 31 it is unclear what is going on here. This confusion needs to be cleared up. This short lifetime appears again on line 155.

Reply: We understand the concerns raised by the reviewer regarding the rapid PL lifetime observed. The reason for such a swift excited-state lifetime is attributed to the excellent face-to-face arrangement of Pt(fprpz)₂ in thin films, which fosters favorable heavy atom Pt-Pt interactions, leading to metal-to-metal-to-ligand charge transfer (MMLCT) transitions that is virtually optically allowed. This viewpoint is supported by the literature published in Nature Photonics (15, 230–237, 2021). Additionally, a similar phenomenon was observed in our previously published article in Advanced Materials (28, 2526–2532, 2016), where Pt(fppz)₂ exhibited an emission at 625 nm with a PLQY of up to 96%, yet the PL lifetime was measured to be as short as 382 ns. Furthermore, the Pt complexes reported in Advanced Functional Materials (30, 2002173, 2020), with strong emission ranging between 776–832nm, had PL lifetimes of only 520–790 ns. Lastly, another Pt complex molecule (named as DR), emitting at 995 nm with a PLQY of 13.3% , had a PL lifetime of merely 30 ns (Nature Photonics 2022, 16, 843–850).

5. Line 138 "Note that the standard deviation of lifetime under an average of four replica gave an error bar of \pm 1.2 ns." What does this mean?

Reply: We apologize for the awkward wording. Since we have attached error values to each piece of data separately in this revision, we rewrite this sentence as follows: "Note that four samples were prepared for each experimental condition and measured separately to obtain an average value and error."

6. Lines 222 -228: The statement "Based on the SCLC results, we can infer that the majority of emission occurs in BTP-eC9. This is attributed to the matched electron mobility, allowing for efficient electron transfer to BTP-eC9. Although there is a significant difference in hole mobility, the lower HOMO level of Pt(fprpz)₂ by 0.4 eV

compared to BTP-eC9 ensures that holes still migrate towards BTP-eC9 under an applied bias." is very confusing. I had thought that the mechanism for exciting the BTP-eC9 was exciton transfer from the Pt complex. Yet this statement clearly states that the emission is attributed to electron transfer to the fluorophore. Which is it? Electron or exciton transfer? It cannot be both. I am afraid the authors are actually unclear as to what is happening in their device.

Reply: We appreciate the reviewer's comments and have accordingly made revisions to the manuscript incorporating the results from SCLC measurements, energy level alignments, and Setfos simulations with an addition of statement below.

“Despite the differences in mobility between Pt(fprpz)₂ and BTP-eC9, our simulations with Setfos have revealed that the alignment of energy levels between the energy donor and acceptor is more crucial than their mobility (see Figure S9 and S10).”

Figure S9. The simulation results using Setfos: Recombination distribution in devices containing different HOMO energy levels of the Pt-complexes.

Figure S10. Recombination distribution in devices containing different electron mobility of the Pt-complexes.

We would like to clarify here that the hole mobility of both NPB and mCP exceeds $10^{-4} \text{ cm}^2 \text{ V}^{-1} \text{ s}^{-1}$. However, a sudden decrease to $3.95 \times 10^{-6} \text{ cm}^2 \text{ V}^{-1} \text{ s}^{-1}$ in $\text{Pt}(\text{fprpz})_2$ leads to a significant accumulation of holes. Likewise, considerable accumulation of electrons occurs at the boundary of $\text{Pt}(\text{fprpz})_2/\text{BTP-eC9}$ for the same reason. Ultimately, this results in extensive recombination of electrons and holes at the interface, which was also confirmed through the Setfos software simulation.

In addition, in this study, we have considered the mobility of electrons and holes in each layer of material. Subsequently, we adjusted the recombination area of electrons and holes based on the energy level difference. The purpose of this optimization is to bring the recombination region appearing in the $\text{Pt}(\text{fprpz})_2$ layer and as close as possible to the BTP-eC9 layer for conducting the energy transfer. Appropriate energy level differences can help us adjust the position of the recombination region. Attached below (Figure S9) is the analysis of the recombination region simulated by Setfos software. Evidently, when HOMO of $\text{Pt}(\text{fprpz})_2$ is lower than that of BTP-eC9 by 0.4 eV, it helps the recombination region shift closer to the $\text{Pt}(\text{fprpz})_2/\text{BTP-eC9}$ interface.

We would like to draw the reviewer's attention that given the complexity and their integral role in achieving efficient energy transfer and overall device efficiency, our responses to the reviewer might not cover all aspects comprehensively. We hope the

reviewers can appreciate the intricacies involved in our research and understand the limitations that might arise in conveying the full scope of our findings and challenges.

7. Line 252: Here the losses are clearly significant. Only a fraction of the excitations on Pt end up on BTP-eC9. Nonradiative decay seems large once again, contradicting the 80% PLQY and fast time constant for triplet decay on the Pt complex.

Reply: We are sorry for the unclear statement that causes the confusion. In this section, we mainly compare the change of Pt(fprpz)₂ layer from 5 nm to 10 nm. Therefore, we made the revision as follows:

“This was also supported by the experiments where the Pt(fprpz)₂ (5 nm)/BTP-eC9 stamped structure exhibited an increased radiance from 31.73 to 34.14 W sr⁻¹ m⁻² and a slightly improved EQE from 2.00 (1.80 ± 0.14%) to 2.07% (1.83 ± 0.15%), accompanied by the reduction of Pt(fprpz)₂'s emission from 11.2% to 4.3%.”

In addition, most of the excited Pt(fprpz)₂ have been transferred to BTP-eC9. The reason why the loss appears to be large at first glance is because the PLQY of BTP-eC9 itself is only 5.57% (5.42 ± 0.17%). This substantial energy transfer enabled the device made of BTP-eC9 fluorescent molecules with PLQY = 5.57% (5.42 ± 0.17%) (improved to 8.85% (8.61 ± 0.17%) by stamping method) to achieve a substantial enhancement from 0.18% (0.14 ± 0.04%) to 2.07% (1.83 ± 0.15%) for the 928 nm device.

There seems to be significant confusion as to mechanisms concluded in this paper that need clarification before this is acceptable.

Reviewer #4 (Remarks to the Author):

The authors have addressed my concerns in the revised manuscript. It is now suitable for publication.

REVIEWER COMMENTS

Reviewer #1 (Remarks to the Author):

The authors have addressed my comments and I support publishing this work in Nature Communications.

Extra comments:

- The design rule 3) is the interesting one and novel compared to the established approach.
- The Jablonski diagram is a helpful addition to the manuscript and greatly improves clarity of the mechanism portrayed by the authors.

Reviewer #2 (Remarks to the Author):

The authors considerably improved the manuscript. However, some revisions are still needed, as detailed below:

1. Page 5, Line 155. The “Since” at the beginning of the sentence should be removed or the sentence rephrased.
2. Page 5, Lines 161-164. The sentence “Therefore, in yet another approach, we employed a stamping technique to directly transfer the solid-state BTP-eC9 onto the Pt(fprpz)₂ layer, thereby circumventing any physical contact such as van der Waal force, hydrogen bonding, p-stacking, etc. in between” sounds like a repetition of the previous sentence and should be removed. In fact, there is no need to vaguely refer to van der Waals force, hydrogen bonding, p-stacking here.
3. Page 6, Line 177-180. “This enhancement shows that in addition to the energy transfer from Pt(fprpz)₂ to the BTP-eC9 pathway, there are other channels, such as the reabsorption of Pt(fprpz)₂ emission by BTP-eC9. In addition, the arrangement of BTP-eC9 may be affected by Pt(fprpz)₂ after being stamped on Pt(fprpz)₂, thereby improving its photophysical properties and PLQY”. While I accept the hypothesis of improved molecular arrangement being a possible reason for the increased PLQY of BTP-eC when stamped on the Pt(fprpz)₂ complex, I still do not understand how “reabsorption” (I suppose authors mean radiative transfer) can play a role too. I appreciate that

authors updated Table S1 by including the PLQY of the bilayer when excited at 505 nm. And even in this case, authors see an increase in PLQY in the acceptor emission range (PLQY up to 7.1% from the initial ~5.5% of neat BTP-eC on glass), which is remarkable. But I think that the only plausible reason for the PLQY increase (other than the one related to optimal molecular packing suggested by the authors) is that the red tail of the Pt complex emission overlaps with the emission of BTP-eC (Fig. S1), therefore contributing to the overall emission in that specific spectral range (> 850 nm). I do not think that radiative transfer or resonant energy transfer should contribute to the increased PLQY of BTP-eC, since the efficiencies of both processes, however high they can be, cannot exceed the unit value.

In conclusion, it is trivial to show that in the case of selective excitation of the acceptor BTP-eC9, radiative or energy transfer cannot be considered as sources of increased PLQY. But even in the case of Pt(fprpz)₂ excitation at 505 nm, I expect the PLQY of the acceptor in the bilayer to be the product of the intrinsic BTP-eC PLQY times the energy transfer and/or the radiative transfer efficiency. Therefore, the PLQY of BTP-eC9 in the bilayer should be inevitably lower than the one of the bare BTP-eC9 on glass, unless the intrinsic PLQY of BTP-eC9 increases upon stamping on Pt(fprpz)₂, for instance for the packing arrangement considerations suggested by the authors.

REVIEWER COMMENTS

Reviewer #1 (Remarks to the Author):

The authors have addressed my comments and I support publishing this work in Nature Communications.

Extra comments:

- The design rule 3) is the interesting one and novel compared to the established approach.
- The Jablonski diagram is a helpful addition to the manuscript and greatly improves clarity of the mechanism portrayed by the authors.

Reply: We are grateful to the reviewer for his/her compliment.

Reviewer #2 (Remarks to the Author):

The authors considerably improved the manuscript. However, some revisions are still needed, as detailed below:

1. Page 5, Line 155. The “Since” at the beginning of the sentence should be removed or the sentence rephrased.

Reply: Thank you very much for the correction, We have removed 'Since' from the manuscript.

2. Page 5, Lines 161-164. The sentence “Therefore, in yet another approach, we employed a stamping technique to directly transfer the solid-state BTP-eC9 onto the Pt(fprpz)₂ layer, thereby circumventing any physical contact such as van der Waal force, hydrogen bonding, p-stacking, etc. in between” sounds like a repetition of the previous sentence and should be removed. In fact, there is no need to vaguely refer to van der Waals force, hydrogen bonding, p-stacking here.

Reply: We fully agree with the reviewer's perspective and have deleted the sentence accordingly.

3. Page 6, Line 177-180. “This enhancement shows that in addition to the energy transfer from Pt(fprpz)₂ to the BTP-eC9 pathway, there are other channels, such as the reabsorption of Pt(fprpz)₂ emission by BTP-eC9. In addition, the arrangement of BTP-eC9 may be affected by Pt(fprpz)₂ after being stamped on Pt(fprpz)₂, thereby improving its photophysical properties and PLQY”. While I accept the hypothesis of

improved molecular arrangement being a possible reason for the increased PLQY of BTP-eC9 when stamped on the Pt(fprzp)₂ complex, I still do not understand how “reabsorption” (I suppose authors mean radiative transfer) can play a role too. I appreciate that authors updated Table S1 by including the PLQY of the bilayer when excited at 505 nm. And even in this case, authors see an increase in PLQY in the acceptor emission range (PLQY up to 7.1% from the initial ~5.5% of neat BTP-eC on glass), which is remarkable. But I think that the only plausible reason for the PLQY increase (other than the one related to optimal molecular packing suggested by the authors) is that the red tail of the Pt complex emission overlaps with the emission of BTP-eC (Fig. S1), therefore contributing to the overall emission in that specific spectral range (> 850 nm). I do not think that radiative transfer or resonant energy transfer should contribute to the increased PLQY of BTP-eC9, since the efficiencies of both processes, however high they can be, cannot exceed the unit value.

In conclusion, it is trivial to show that in the case of selective excitation of the acceptor BTP-eC9, radiative or energy transfer cannot be considered as sources of increased PLQY. But even in the case of Pt(fprzp)₂ excitation at 505 nm, I expect the PLQY of the acceptor in the bilayer to be the product of the intrinsic BTP-eC9 PLQY times the energy transfer and/or the radiative transfer efficiency. Therefore, the PLQY of BTP-eC9 in the bilayer should be inevitably lower than the one of the bare BTP-eC9 on glass, unless the intrinsic PLQY of BTP-eC9 increases upon stamping on Pt(fprzp)₂, for instance for the packing arrangement considerations suggested by the authors.

Reply: Regarding the concerns raised by the reviewer, we fully understand and apologize once again for the imprecision in our textual descriptions. We attribute the increase in PLQY to the reduction in d-spacing of BTP-eC9. Accordingly, the corresponding statement in this revised version has been rewritten as follows.

“The results clearly indicate that the arrangement of BTP-eC9, after stamped on Pt(fprzp)₂, was affected to shorten the d-spacing, thereby improving its face-on orientation^[42] and hence PLQY. Relevant steady-state PL, lifetime, and PLQY data based on the stamping method are summarized in Table S1 of supporting information.”

We hope that the above replies and corresponding revisions are satisfactory. Thank you very much for the kind assistance.

Sincerely,

Pi-Tai Chou

National Chair Professor